# Code2Worlds: Empowering Coding LLMs for 4D World Generation

**Yi Zhang** [1 2 *]   **Yunshuang Wang** [1 3 *]   **Zeyu Zhang** [1 * †]   **Hao Tang** [1 ‡]

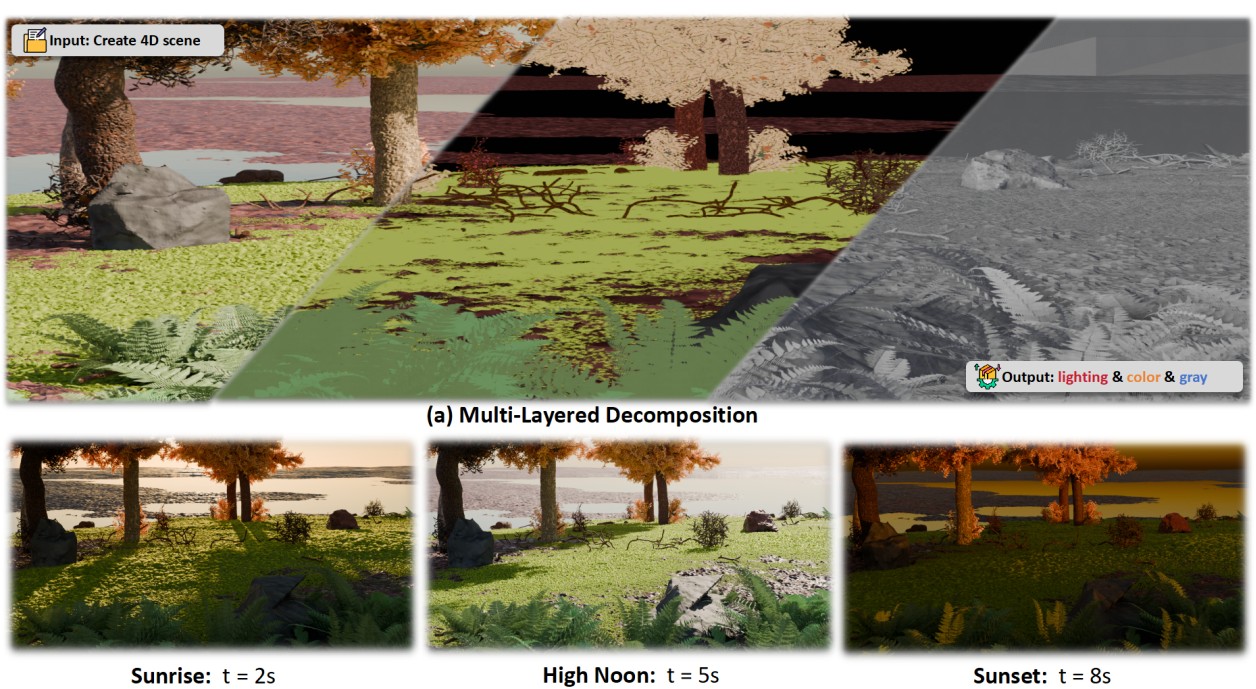

**(a) Multi-Layered Decomposition**

Input: Create 4D scene

Output: lighting & color & gray

Sunrise: t = 2s          High Noon: t = 5s          Sunset: t = 8s
**(b) A 10 second time lapse of a summer forest: sunrise to hign noon to sunset.**

*Figure 1.* **Overview of Code2Worlds. Top:** Decomposition of intrinsic attributes (lighting, color, gray). **Bottom:** A summer forest time-lapse demonstrating coherent atmospheric evolution. The sequence tracks precise transitions from sunrise ($t = 2s$) to noon ($t = 5s$) and sunset ($t = 8s$).

## Abstract

Achieving spatial intelligence requires moving beyond visual plausibility to build world simulators grounded in physical laws. While coding LLMs have advanced static 3D scene generation, extending this paradigm to 4D dynamics remains a critical frontier. This task presents two fundamental challenges: multi-scale context entanglement, where monolithic generation fails to balance local object structures with global environmental layouts; and a semantic-physical execution gap, where open-loop code generation leads to physical hallucinations lacking dynamic fidelity. We introduce **Code2Worlds**, a framework that formulates 4D generation as language-to-simulation code generation. First, we propose a dual-stream architecture that disentangles retrieval-augmented object generation from hierarchical environmental orchestration. Second, to ensure dynamic fidelity, we establish a physics-aware closed-loop mechanism in which a Post-Process Agent scripts dynamics, coupled with a VLM-Motion Critic that performs self-reflection to iteratively refine simulation code. Evaluations on the Code4D benchmark show Code2Worlds outperforms baselines with a 41% SGS gain

[*]Equal contribution. [†]Project lead. [1]School of Computer Science, Peking University [2]Southwest University [3] Institute of Information Engineering, Chinese Academy of Sciences. [‡]Correspondence to: Hao Tang <bjdxtanghao@gmail.com>.

*Proceedings of the 43rd International Conference on Machine Learning*, Seoul, South Korea. PMLR 306, 2026. Copyright 2026 by the author(s).

and 49% higher Richness, while uniquely generating physics-aware dynamics absent in prior static methods. Code: https://github.com/AIGeeksGroup/Code2Worlds. Website: https://aigeeksgroup.github.io/Code2Worlds.

## 1. Introduction

While generative models have advanced from static images to high-fidelity videos (Nikankin et al., 2023; Lee et al., 2024), achieving true spatial intelligence requires world simulators grounded in causal physical laws rather than superficial pixel dynamics. In this pursuit, procedural code generation has emerged as a uniquely powerful paradigm. Unlike black-box neural representations, executable programs offer rigorous control over 3D scene structure and semantics, having already demonstrated remarkable success in generating diverse and high-fidelity static 3D scenes (Raistrick et al., 2023; Sun et al., 2025).

However, advancing from static 3D scenes to physically grounded 4D environments via code generation encounters two significant challenges: First, monolithic methods struggle with multi-scale context entanglement. The task of world generation requires resolving conflicting objectives across different scales simultaneously. This involves generating intricate local 3D structures, such as the detailed cortex of a tree, while orchestrating global environments, including atmospheric lighting and terrain layout. A single generation pass often fails to balance these disparate granularities, prioritizing global coherence over local structural intricacy. This trade-off frequently yields target objects with coarse 3D structures that are ill-suited for fine-grained physical actuation, thereby limiting the realism and plausibility of subsequent dynamic simulations.

Second, the transition from static 3D structures to physical dynamics reveals a fundamental execution gap. Prior code-to-scene methods are limited to static appearances and lack temporal simulation capabilities. Extending this paradigm to 4D requires translating abstract semantic motion descriptors, such as leaves trembling, into precise simulation parameters, such as vertex weights and turbulence force fields. This process is currently an open-loop endeavor where coding large language models (LLMs) is like being a blind engineer without visual feedback. This disconnect frequently leads to physical hallucinations where generated motions are syntactically valid but violate basic laws of physics, for instance, rigid bodies distorting or particles ignoring gravity. This results in a severe misalignment between semantic instructions and the actual temporal simulation.

To address these challenges, we introduce *Code2Worlds*, a *Language-to-Simulation* framework. First, to resolve multi-scale entanglement, we propose a dual-stream architecture that decouples the target object from the environmental background. This ensures the focal object acquires the rich structural details. Second, to bridge the execution gap, we establish a closed-loop refinement mechanism driven by a *PostProcess Agent* and self-reflection. Instead of blind open-loop scripting, our system employs a *VLM-Motion Critic* to evaluate rendered dynamics, iteratively correcting physical hallucinations to ensure alignment with user intent.

Our specific contributions are threefold:

- **Factorized Language-to-Simulation Framework.** We propose a dual-stream architecture that disentangles complex scene generation into retrieval-augmented object generation and hierarchical environmental orchestration. This factorization resolves multi-scale context entanglement, ensuring that target objects possess the high-fidelity structural intricacy.

- **Physics-Aware Closed-Loop Correction.** We introduce a two-stage mechanism where a *PostProcess Agent* actuates static scenes via coding dynamic, and a *VLM-Motion Critic* enables self-reflection. This closed-loop system iteratively evaluates rendered rollouts to mitigate temporal artifacts and enforce physical consistency, bridging the semantic-physical gap.

- **Code4D Benchmark and Evaluation.** We construct *Code4D*, a comprehensive benchmark designed to evaluate 4D scene generation. Extensive experiments demonstrate that Code2Worlds consistently outperforms prior code-to-scene frameworks, achieving a 41% improvement in SGS and a 49% increase in Richness, while uniquely generating physics-aware dynamics that prior static methods lack.

**Conflict of Interest Disclosure.** The authors declare that they have no financial conflicts of interest related to this work.

## 2. Related Work

**3D and 4D Content Generation.** Text-driven 3D generation has evolved from early procedural methods (Coyne & Sproat, 2001; Chang et al., 2014) and layout optimization (Fisher et al., 2012; Yu et al., 2011) to modern learning-based approaches like DreamFusion (Poole et al., 2022) and Magic3D (Lin et al., 2023). However, extending these paradigms to 4D scenes faces significant hurdles. While MAV3D (Singer et al., 2023) pioneered text-to-4D generation by optimizing dynamic NeRFs with video diffusion priors, it is hindered by high computational costs and limited editability. Recent Gaussian-based methods DreamGaussian4D (Ren et al., 2023) and SP-GS (Wan et al., 2024) have

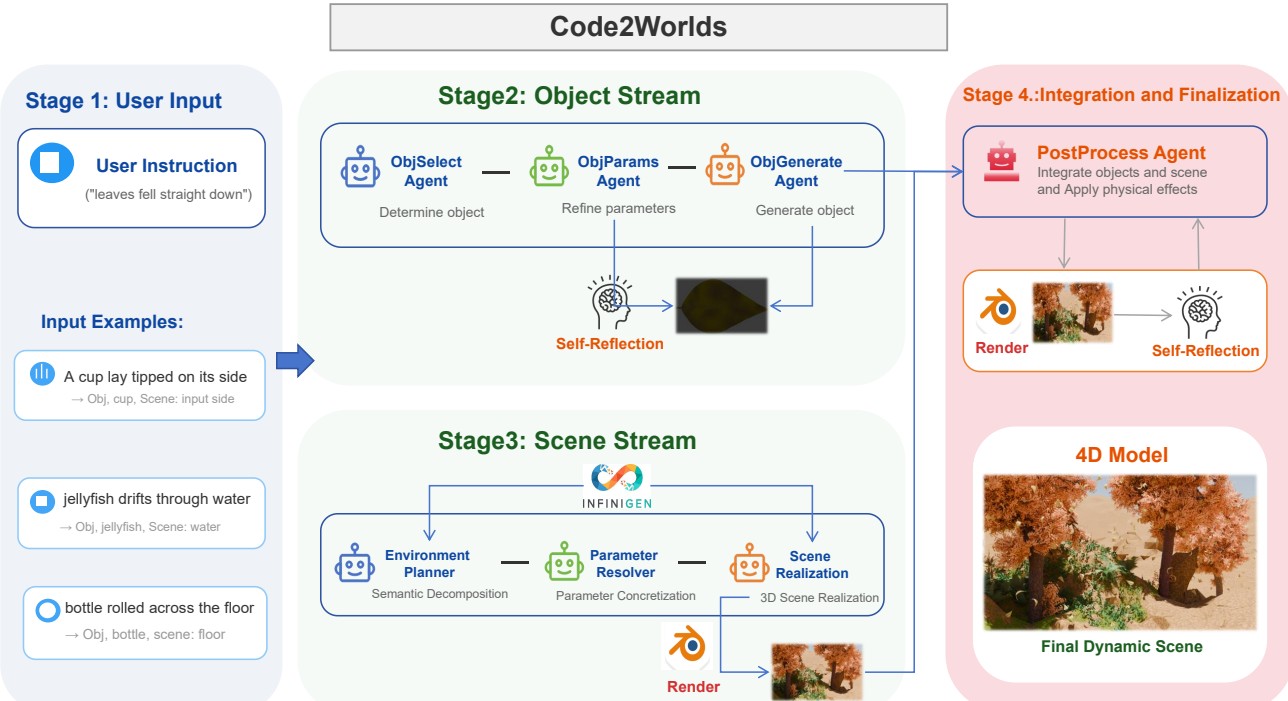

Figure 2. **Code2Worlds Execution Pipeline.** The framework generates 4D scenes via a dual-stream architecture: 1) an *Object Stream* utilizing retrieval augmented parameter generation with object self-reflection; 2) a *Scene Stream* employing hierarchical environmental orchestration; and 3) refinement mechanism driven by a *PostProcess Agent* and self-reflection.

improved efficiency; yet, generating full-scale 4D scenes remains challenging due to physical consistency. Beyond 4D content generation, MWM (Yan et al., 2026) studies action-conditioned prediction for mobile agents, and FO-LIAGE (Liu & Tang, 2025) models physical dynamics via unbounded surface evolution. Code2Worlds instead focuses on language-to-simulation code generation for controllable 4D scenes.

**LLM-driven Procedural Modeling.** The code-to-scene paradigm leverages LLMs to generate executable code for 3D software. Foundational works like 3D-GPT (Sun et al., 2025) and Infinigen (Raistrick et al., 2023) enabled text-to-scene generation, while SceneCraft (Hu et al., 2024) introduced self-improvement mechanisms. To handle complex descriptions, RPG (Yang et al., 2024) employs global planning for task decomposition, a strategy further refined by specialized agents like LL3M (Lu et al., 2025) and VUL-CAN (Kuang et al., 2026). Additionally, recent works incorporate retrieval-augmented generation (RAG) to reduce syntactic errors. Despite this, programmatic modeling remains predominantly optimized for static 3D environments.

**Multi-Agent Coordination and Reflection.** Multi-agent systems have emerged as robust frameworks for collaborative reasoning, outperforming monolithic models in complex task decomposition (Liu et al., 2023; Gong et al., 2024). To

mitigate error propagation in open-loop pipelines, recent work emphasizes closed-loop self-correction, exemplified by LATS (Zhou et al., 2024), which uses feedback for iterative refinement. Unlike methods designed for static or symbolic domains that overlook 4D physical discrepancies, we propose a parallel multi-agent architecture optimized for 4D, leveraging concurrent generation and dual-stream reflection to ensure temporal-physical coherence.

## 3. The Proposed Method

### 3.1. Overview

We introduce Code2Worlds, a framework leveraging coding LLMs to generate 4D scenes from text. As shown in Figure 2, it employs a dual-stream architecture: a Scene Stream for procedural environmental layout and an Object Stream for detailed 3D objects. Finally, a PostProcess Agent integrates these components and scripts temporal dynamics, supervised by a VLM critic to ensure semantic and physical consistency.

### 3.2. Object Stream: Parametric Object Generation

Instead of generating raw 3D structures from scratch, we propose a Retrieval-Augmented Parametric Generation method. This stream capitalizes on the robust procedural priors inherent in the Infinigen procedural generator.

Specifically, it translates semantic instructions into parameter spaces and subsequently generates executable procedural codes to instantiate high-fidelity objects with precise 3D appearance and rich textures.

**Dynamic-Aware Object Selection.** The `ObjSelect Agent` parses the instruction $\mathcal{I}$ to identify entities that require specific dynamic interactions. For instance, if $\mathcal{I}$ involves "leaves drift," the agent isolates "leaf" as a target object. Conversely, global environmental changes (e.g., sunlight shifting) bypass this stream, as they are deferred to the `PostProcess Agent` in Section 3.4 for unified dynamic simulation. Formally, we formulate the target selection as an optimization problem:

$$e_{\text{target}} = \arg \max_{e \in \mathcal{E}(\mathcal{I})} P_{\text{dyn}}(e \mid I), \qquad (1)$$

where we identify the primary subject $e_{\text{target}}$ from candidates $\mathcal{E}(\mathcal{I})$ by maximizing dynamic necessity $P_{\text{dyn}}$.

**Retrieval-Augmented Parameter Generation.** Predicting high-dimensional procedural parameters directly is challenging for LLMs due to the lack of domain-specific priors. To bridge this gap, we construct a Procedural Parameters Library $\mathcal{L}_{param}$, which summarizes Infinigen's complicated parameters into structured schema documents in Section E. Once a target dynamic entity $e_{\text{target}}$ is selected by the `ObjSelect Agent`, the system queries $\mathcal{L}_{\text{param}}$ to retrieve its specific parameter definition $\mathcal{S}_{\text{ref}}$.

$$\mathcal{S}_{\text{ref}} \leftarrow \texttt{Retrieve}(\mathcal{L}_{\text{param}}, e_{\text{target}}). \qquad (2)$$

As exemplified in the `LeafFactory` schema, this document explicitly disentangles the parameter space into three dimensions: (1) *Structural Shape*, which comprises parameters governing 3D form such as `midrib_length`; (2) *Surface Texture*, capturing detailed appearance attributes like `vein_density`; and (3) *Material Semantics*, defining rendering properties including `blade_color_hsv`.

Crucially, we augment these definitions with semantic exemplars paired with demonstrations of natural-language descriptions and their corresponding ground-truth parameter configurations in Section E. These exemplars provide concrete mappings from qualitative descriptions to precise quantitative values, enabling the LLM to infer complex parameter combinations via in-context analogical reasoning. Consequently, the agent generates $\mathcal{S}$ by conditioning on the retrieved reference $\mathcal{S}_{\text{ref}}$ and feedback $\mathcal{F}_{\text{obj}}$ from `Object Self-Reflection` in Section 3.2. Formally, this generation process is defined as:

$$\mathcal{S} \leftarrow \texttt{ObjParam}(\mathcal{S}_{\text{ref}}, \mathcal{I}, \mathcal{F}_{\text{obj}}). \qquad (3)$$

**Semantic-to-Parametric Mapping.** While Section 3.2 generates the parameters, translating these into executable

code presents a twofold challenge: (1) *Syntactic Complexity*, as LLMs lack Infinigen's specific templates; and (2) *Structural Constraints*, which mandate passing parameters as factory arguments rather than variable assignments. To address these challenges, we adopt a retrieval-based strategy mirroring the `ObjParam Agent`. We construct a Reference Code Library $\mathcal{L}_{\text{code}}$, which indexes verified canonical implementations for object categories. Upon selecting the target object, the system queries $\mathcal{L}_{\text{code}}$ to retrieve a reference implementation $\mathcal{C}_{\text{ref}}$.

$$\mathcal{C}_{\text{ref}} \leftarrow \texttt{Retrieve}(\mathcal{L}_{\text{code}}, e_{\text{target}}). \qquad (4)$$

The `ObjGenerate Agent` then generates the final code $\mathcal{C}$. Specifically, the LLM generates $\mathcal{C}$ by integrating the parameters $\mathcal{S}$ and $\mathcal{C}_{\text{ref}}$, which is subsequently executed to instantiate fine-grained 3D object $\mathcal{C}_{\text{obj}}$. Formally, we denote this object generation process as:

$$\mathcal{C}_{\text{obj}} \leftarrow \texttt{ObjGenerate}(\mathcal{C}_{\text{ref}}, \mathcal{S}). \qquad (5)$$

**Object Self-Reflection.** To ensure visual alignment, we introduce a closed-loop reflection mechanism. Upon generating the object, the system renders a 2D snapshot $V_{\text{img}}$ of the object. This visual output, along with the original instruction $\mathcal{I}$, is fed into a VLM that serves as a semantic critic. The VLM assesses the alignment between the rendered appearance and the description to generate signal $\mathcal{V}$. If the object satisfies the requirements, it is validated for scene integration. Conversely, the VLM generates constructive natural language feedback $\mathcal{F}_{\text{obj}}$ for refinement. Formally, this is denoted as:

$$\mathcal{F}_{\text{obj}}, \mathcal{V} \leftarrow \texttt{VLM-Critic}(V_{\text{img}}, \mathcal{I}).$$

Crucially, this feedback is propagated back to `ObjParam Agent`, triggering a regeneration cycle where the agent adjusts specific parameters based on $\mathcal{F}_{\text{obj}}$. This iterative process ensures semantic alignment between the 3D structure and the visual intent.

### 3.3. Scene Stream: Hierarchical Environmental Orchestration

Complementing the Object Stream, the Scene Stream orchestrates the global environment. To address hyperparameter entanglement, we employ a hierarchical method using a structured execution manifest to decouple planning from execution. This drives a three-stage pipeline progressively concretizing abstract intent into rigorous procedural constraints, yielding a coherent 3D environment.

**Semantic Decomposition.** Natural language instructions are inherently sparse and underspecified. A user may simply request a forest, but a complete 3D world requires exhaustive definitions of season, terrain, and vegetation density.

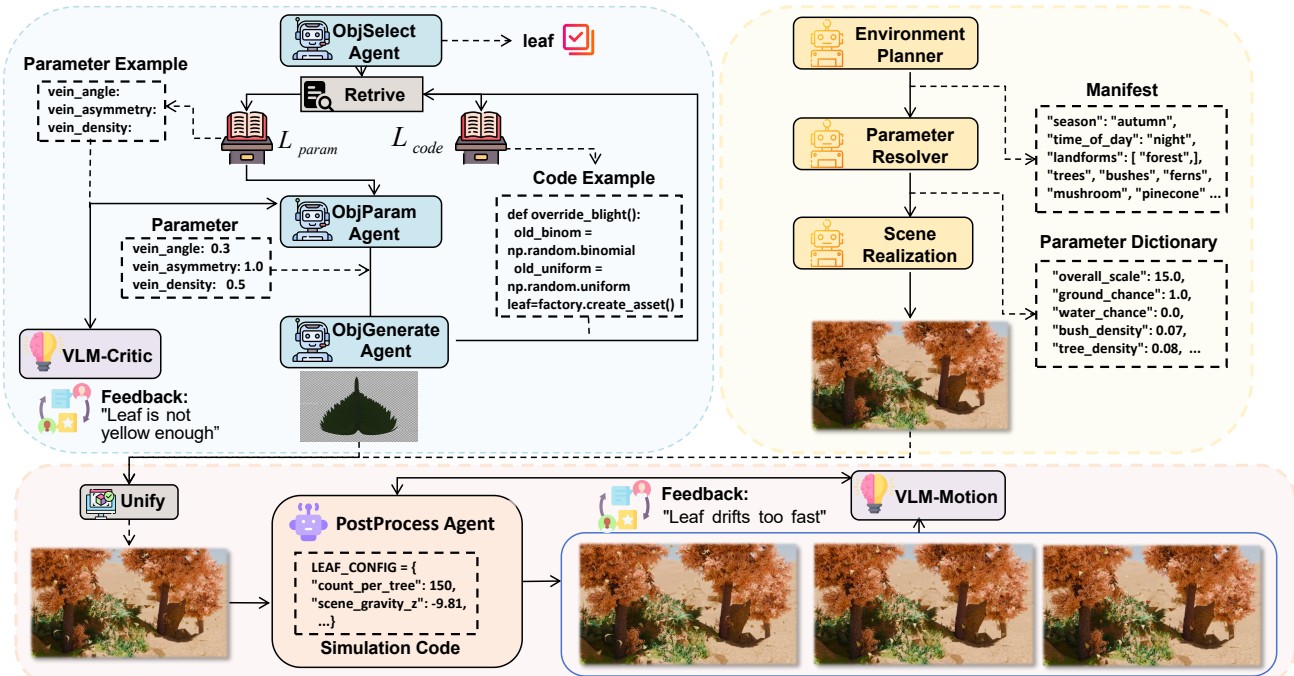

*Figure 3.* A detailed workflow for generating a 4D scene, integrating environmental scene, object generation, and feedback-driven refinement to ensure realistic scene rendering.

The `Environment Planner` acts as a creative extrapolation brain. Its primary role is to bridge the information asymmetry between the sparse instruction and the dense reality of a scene by inferring latent environmental context based on the LLM's intrinsic world knowledge. Formally, given the instruction $\mathcal{I}$, it decomposes instruction $\mathcal{I}$ into a manifest $\mathcal{M}$:

$$\mathcal{M} \leftarrow \texttt{Planner}(\mathcal{I}). \qquad (6)$$

Rather than relying on superficial keyword matching, the agent infers latent environmental variables to construct a comprehensive environmental specification across three dimensions: (1) *Atmospheric Context*, where it infers the environmental atmosphere. For example, given "spooky forest", it infers an "Autumn" season with "heavy fog" and "dim lighting" to align with the stylistic intent; (2) *Terrain Morphology*, enforcing geomorphological consistency by autonomously instantiating implied features like rivers even if not explicitly requested; and (3) *Vegetation Density*, enriching the ecosystem with coherent understory elements like bushes. This generative inference transforms sparse instructions into holistic, environmentally rich specifications.

**Parameter Concretization.** Whereas Section 3.3 determines categorical existence, the `Parameter Resolver` governs parametric magnitude. Its primary objective is to address scale ambiguity by grounding qualitative semantic descriptors into precise scalars. For instance, consider a directive for a "dense forest". The agent translates abstract intensity into specific biological densities, such as

setting `max_tree_species` to 5 to maximize biodiversity. Formally, given the manifest $\mathcal{M}$, the `Parameter Resolver` infers the scene parameter dictionary $\mathcal{D}$:

$$\mathcal{D} \leftarrow \texttt{Resolver}(\mathcal{M}). \qquad (7)$$

Beyond simple mapping, this agent enforces logical consistency across the parameter space to prevent semantic and physical conflicts. Specifically, if the inferred environment is "rainforest," the agent explicitly prunes incompatible objects by forcing `snow_layer_chance` to zero. Furthermore, it resolves parameter couplings between variables. It ensures that coupled parameters, such as `air_density` and `dust_density`, are jointly calibrated to mitigate rendering anomalies, which often arise from incoherent physical properties.

**3D Scene Realization.** The final phase executes the transition from parametric specifications to 3D environments. The `Scene Realization` operates as a dual-stage synthesizer. First, acting as a domain-specific compiler, it converts the resolved parameter dictionary into valid execution code compatible with Infinigen's internal schema. It systematically maps high-level logical flags, such as `terrain.ground`, to primitives, such as `scene.ground_chance`. Formally, given the scene parameter dictionary $\mathcal{D}$, `Scene Realization` generates the 3D scene $\mathcal{C}_{\text{env}}$:

$$\mathcal{C}_{\text{env}} \leftarrow \texttt{Realizer}(\mathcal{D}). \qquad (8)$$

| Method | Text Control | Static Layout | Object Details | Dynamics | Temporal | Self-Reflect |
|---|:---:|:---:|:---:|:---:|:---:|:---:|
| MeshCoder (Dai et al., 2026) | ✓ | × | ✓ | × | N/A | ✓ |
| Infinigen (Raistrick et al., 2023) | × | ✓ | ✓ | × | N/A | × |
| Infinigen Indoors (Raistrick et al., 2024) | × | ✓ | ✓ | × | N/A | × |
| 3D-GPT (Sun et al., 2025) | ✓ | ✓ | △ | × | N/A | × |
| SceneCraft (Hu et al., 2024) | ✓ | ✓ | △ | × | N/A | ✓ |
| ImmerseGen (Yuan et al., 2026) | ✓ | ✓ | △ | △ | △ | △ |
| **Code2Worlds (Ours)** | ✓ | ✓ | ✓ | ✓ | ✓ | ✓ |

*Table 1.* **We compare methods under the Code4D criteria.** Text Control: generation conditioned on natural language prompts; Static Layout: global scene arrangement capability; Object Details: fine-grained geometry and texture quality; Dynamics: instruction-grounded physics-aware 4D dynamics; Temporal: temporal consistency in rendered rollouts; Self-Reflection: iterative visual self-correction. Symbols: ✓ supported; △ partially supported; × not supported; N/A not applicable.

Crucially, the execution code serves merely as the intermediate instruction set. The agent subsequently invokes the Infinigen program to execute these codes, instantiating the 3D scene procedurally. This structured interface decouples semantic planning from code execution. It ensures strict adherence to upstream constraints while eliminating the syntactic instability inherent in free-form code generation.

### 3.4. Physics-Aware 4D Scene Generation

The terminal phase generates a cohesive 4D scene by integrating discrete objects with the global scene under the governance of the `PostProcess Agent`. The `PostProcess Agent` acts as a physics engine, translating kinetic cues into simulation constraints to animate the scene. Crucially, we employ *Dynamic Effects Self-Reflection*, where a VLM evaluates video rollouts and iteratively calibrates parameters to ensure semantic coherence.

**Dynamic Scene Integration.** The agent first initiates the pipeline by unifying the object from Section 3.2 with the global scene generated by Section 3.3. It then generates a comprehensive Blender script to actuate the scene based on the user's instructions and feedback $\mathcal{F}_{\text{dyn}}$ from Section 3.4. This process involves two critical steps: (1) parameter inference grounds qualitative descriptions into quantitative parameters, such as mapping "peacefully" to a `wind_strength` coefficient of 0.25; and (2) procedural actuation generates code to apply gradient masks for structural deformation, anchoring tree roots while allowing branches to sway. Crucially, this step enforces collision constraints, ensuring particle-terrain interactions. Formally, this two-stage scene realization is defined as:

$$\mathcal{P}_{\text{phys}} \leftarrow \text{INFERPHYSICS}(\mathcal{I}, \mathcal{F}_{\text{dyn}}), \qquad (9)$$

$$\mathcal{W}_{\text{dyn}} \leftarrow \text{ACTUATE}(\mathcal{W}_{\text{static}}, \mathcal{P}_{\text{phys}}), \qquad (10)$$

where $\mathcal{I}$ and $\mathcal{F}_{\text{dyn}}$ denote the user instruction and dynamic feedback, respectively. The function `InferPhysics` maps these inputs to quantitative physics parameters $\mathcal{P}_{\text{phys}}$. Subsequently, ACTUATE applies these physics constraints

to the unified static geometry $\mathcal{W}_{\text{static}}$, resulting in the final dynamic scene $\mathcal{W}_{\text{dyn}}$.

**Dynamic Effects Self-Reflection.** To ensure the semantic alignment of generated dynamics, we extend the self-reflection mechanism from the spatial domain to the temporal domain. The system renders a video rollout, which is then fed into a VLM that acts as a motion critic. The VLM evaluates whether the rendered dynamic effects match the instruction $\mathcal{I}$. For instance, if the instruction specifies a "gentle breeze" but the rendered footage shows trees thrashing violently, the VLM identifies this discrepancy as a magnitude error. This feedback drives a closed-loop refinement cycle within Section 3.4, iteratively calibrating physics hyperparameters to ensure the final 4D scene exhibits both structural integrity and temporal semantic coherence. Formally, this temporal reflection mechanism is denoted as:

$$\mathcal{F}_{\text{dyn}}, \text{valid} \leftarrow \text{VLM-MOTION}(V_{\text{video}}, \mathcal{I}), \qquad (11)$$

where $V_{\text{video}}$ is the rendered video rollout of the generated dynamic scene $\mathcal{W}_{\text{dyn}}$. The function VLM-MOTION compares this footage against the original instruction $\mathcal{I}$ to produce constructive feedback $\mathcal{F}_{\text{dyn}}$ and a boolean validation signal valid for the refinement loop.

## 4. Experiments

We evaluate Code2Worlds via comprehensive quantitative metrics, with qualitative demonstrations detailed in Section F.

### 4.1. Benchmark and Metrics

**Code4D Benchmark.** The Code4D Benchmark accesses the framework across three dimensions: object, scene, and dynamic generation. We evaluate our approach against both state-of-the-art code-centric methods and leading text-to-video generation models. More details are provided in Section D.

| Method | Object Generation | | | Scene Generation | | | |
|---|---|---|---|---|---|---|---|
| | O-CLIP ↑ | SGS ↑ | Style-CLIP ↑ | S-CLIP ↑ | Failure Rate ↓ | HRS ↑ | Richness ↑ |
| MeshCoder (Dai et al., 2026) | 0.2027 | 14.6 | 0.6406 | – | – | – | – |
| Infinigen (Raistrick et al., 2023; 2024) | 0.2431 | 35.5 | 0.6671 | 0.2113 | × | × | 41.0 |
| 3D-GPT (Sun et al., 2025) | 0.2075 | 37.0 | 0.6178 | 0.1737 | × | × | 41.7 |
| SceneCraft* (Hu et al., 2024) | 0.2411 | 34.6 | 0.6490 | 0.2384 | × | × | 15.2 |
| ImmerseGen* (Yuan et al., 2026) | 0.2417 | 43.5 | 0.5991 | 0.2210 | × | × | 35.5 |
| **Code2Worlds (Ours)** | **0.2655** | **61.4** | **0.6734** | **0.2432** | **10%** | **55.4** | **62.3** |

| Setting | Motion Smoothness ↑ | Subject Consistency ↑ | Failure Rate ↓ | Background Consistency ↑ | Temporal Flickering ↑ |
|---|---|---|---|---|---|
| Stable Video Diffusion (Blattmann et al., 2023) | 0.9913 | 0.9312 | 50% | 0.9702 | 0.9859 |
| Animatediff (Guo et al., 2024) | 0.9833 | **0.9778** | 70% | **0.9746** | 0.9743 |
| CogVideoX (Yang et al., 2025) | 0.9912 | 0.9004 | 50% | 0.9463 | 0.9893 |
| Hunyuan (Kong et al., 2024) | 0.9925 | 0.9406 | 30% | 0.9717 | 0.9899 |
| **Code2Worlds (Ours)** | **0.9952** | 0.9415 | **10%** | 0.9710 | **0.9949** |

*Table 2.* The top panel compares our framework with code-centric methods in static object and scene generation, while the bottom panel provides a comparative analysis against video diffusion models. Methods marked with * are reproduced by us.

**Evaluation Metrics.** We adopt a multidimensional evaluation protocol comprising three metric categories. First, we use CLIP-based (Radford et al., 2021) scores to assess semantic and stylistic alignment: O-CLIP and S-CLIP evaluate consistency across object and static scene dimensions, while Style-CLIP quantifies contextual compatibility by computing the visual similarity between the generated object and the target scene image. Second, we utilize VBench (Huang et al., 2024) to quantify temporal coherence and video stability via Motion Smoothness, Subject Consistency, Background Consistency, and Temporal Flickering. Third, we leverage GPT-4o (Hurst et al., 2024) to assess perceptual fidelity through metrics such as SGS for fine-grained object attributes, Richness for environmental complexity, and HRS for visual-physical plausibility. Finally, we report a physics Failure Rate derived from manual inspection to quantify objective simulation violations, including rigid-body interpenetration, unnatural gravitational detachment, and collision mishandling.

### 4.2. Main Results

**Capability Analysis.** As shown in Table 1, existing methods exhibit limitations: the Infinigen family (Raistrick et al., 2023; 2024; Joshi et al., 2025) excels in object detail but lacks natural language control and reflection; conversely, text-driven approaches like MeshCoder, 3D-GPT, and SceneCraft are restricted to static scenes. While ImmerseGen attempts to model dynamics, it lacks temporal consistency. Code2Worlds uniquely bridges these gaps by unifying text controllability, high-fidelity layouts, and physics-aware 4D dynamics via iterative self-reflection.

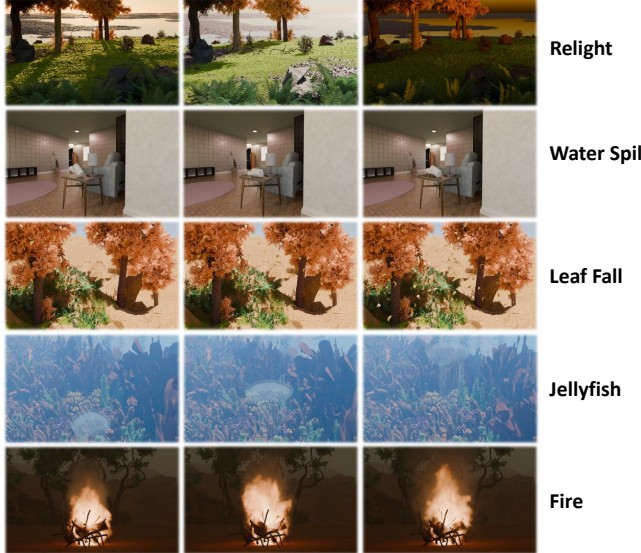

*Figure 4.* A series of environmental effects rendered in different scenes: 1) Relighting adjustments, 2) Water spill interaction, 3) Leaf fall simulation, 4) Jellyfish movement in an aquatic environment, and 5) Fire effect in a natural setting.

**Comparison on Object Generation.** We evaluate object generation against procedural, reconstruction-based, and agent-centric baselines, represented by Infinigen, Mesh-Coder, and ImmerseGen, respectively. MeshCoder, which we augment with Point-E for point-cloud-to-script conversion, exhibits limited robustness and the lowest structural fidelity. While agent-based methods such as ImmerseGen achieve improved performance with an SGS of 43.5, they still lack structural detail. In contrast, Code2Worlds establishes a new state-of-the-art across all metrics and achieves

**Algorithm 1** Unified 4D Scene generation Framework
1: **Input:** Natural language instruction $\mathcal{I}$, Libraries $\mathcal{L}_{\text{param}}, \mathcal{L}_{\text{code}}$
2: **Output:** 4D Scene $\mathcal{W}_{\text{4D}}$
3: *// Phase 1: Object Stream*
4: $e_{\text{target}} \leftarrow \arg\max_e P_{\text{dyn}}(e \mid \mathcal{I})$     *# Target Selection*
5: $\mathcal{S}_{\text{ref}}, \mathcal{C}_{\text{ref}} \leftarrow \text{RETRIEVE}(\mathcal{L}_{\text{param}}, \mathcal{L}_{\text{code}}, e^*)$
6: Initialize feedback $\mathcal{F}_{\text{obj}} \leftarrow \emptyset$
7: **repeat**
8:    $\mathcal{S} \leftarrow \text{OBJPARAMS}(\mathcal{S}_{\text{ref}}, \mathcal{I}, \mathcal{F}_{\text{obj}})$
9:    $\mathcal{C}_{\text{obj}} \leftarrow \text{OBJGENERATE}(\mathcal{C}_{\text{ref}}, \mathcal{S})$
10:    $V_{\text{img}} \leftarrow \text{RENDER}(\mathcal{C}_{\text{obj}})$
11:    $\mathcal{F}_{\text{obj}}, \text{valid} \leftarrow \text{VLM-CRITIC}(V_{\text{img}}, \mathcal{I})$
12: **until** valid is true     *# Object Reflection Loop*
13: *// Phase 2: Scene Stream*
14: $\mathcal{M} \leftarrow \text{PLANNER}(\mathcal{I})$     *# Semantic Decomposition*
15: $\mathcal{D} \leftarrow \text{RESOLVER}(\mathcal{M})$     *# Parameter Concretization*
16: $\mathcal{C}_{\text{env}} \leftarrow \text{REALIZER}(\mathcal{D})$     *# 3D Scene Realization*
17: *// Phase 3: 4D Scene Synthesis*
18: $\mathcal{W}_{\text{static}} \leftarrow \text{UNIFY}(\mathcal{C}_{\text{obj}}, \mathcal{C}_{\text{env}})$
19: Initialize feedback $\mathcal{F}_{\text{dyn}} \leftarrow \emptyset$
20: **repeat**
21:    $\mathcal{P}_{\text{phys}} \leftarrow \text{INFERPHYSICS}(\mathcal{I}, \mathcal{F}_{\text{dyn}})$     *# Grounding*
22:    $\mathcal{W}_{\text{dyn}} \leftarrow \text{ACTUATE}(\mathcal{W}_{\text{static}}, \mathcal{P}_{\text{phys}})$
23:    $V_{\text{video}} \leftarrow \text{RENDER}(\mathcal{W}_{\text{dyn}})$
24:    $\mathcal{F}_{\text{dyn}}, \text{valid} \leftarrow \text{VLM-MOTION}(V_{\text{video}}, \mathcal{I})$
25: **until** valid is true     *# Dynamic Reflection Loop*
26: **return** $\mathcal{W}_{\text{4D}} \leftarrow \mathcal{W}_{\text{dyn}}$

| Setting | O-CLIP ↑ | SGS ↑ | Style-CLIP ↑ |
|---|---|---|---|
| w/o $\mathcal{L}_{param}$ | 0.2511 | 48.8 | 0.6535 |
| w/o Retrive | 0.2221 | 23.5 | 0.6578 |
| w/o VLM-Critic | 0.2388 | 58.6 | 0.6591 |
| **Ours** | **0.2655** | **61.4** | **0.6734** |

*Table 3.* **Ablation on object generation components.**

| Setting | O-CLIP ↑ | Failure Rate ↓ | SGS ↑ | HRS ↑ |
|---|---|---|---|---|
| w/o VLM-Critic | 0.2388 | – | 58.6 | – |
| w/o VLM-Motion | – | 60% | – | 47 |
| **Ours** | **0.2655** | **10%** | **61.4** | **55.4** |

*Table 4.* **Ablation on self-reflection mechanisms.**

ity, achieving a Motion Smoothness of 0.9952 and a Temporal Flickering score of 0.9949. These results highlight the inherent stability of our approach, which ensures consistency through deterministic 3D rendering rather than latent-space interpolation, effectively eliminating the stochastic noise common in diffusion-based processes. Conversely, models like AnimateDiff reveal a trade-off between local appearance and global coherence: while they maintain high consistency by freezing pixel identity, they lack a 3D structural representation. This absence results in low motion smoothness and Failure Rates reaching 70%, manifested as texture boiling and physical artifacts during transitions. By grounding generation in executable code, our framework significantly mitigates these violations while preserving textural stability.

### 4.3. Ablation Study

We conduct a component-wise analysis to validate our design choices. More ablation experiments are detailed in Section B.

**Object Generation Components.** In Table 3, we evaluate the object generation pipeline. w/o Retrieve refers to bypassing the retrieval module, causing the LLM to generate the procedural script directly from the prompt. w/o $\mathcal{L}param$ indicates the exclusion of the Structured Procedural Parameters Library, which limits the LLM's knowledge of the specific, controllable parameters available for the asset. The results reveal that omitting retrieval results in the largest performance drop, with an SGS value of 23.5, highlighting the importance of retrieved reference scripts for proper initialization. Moreover, excluding $\mathcal{L}param$ substantially reduces fidelity, demonstrating that exposing a well-defined parameter space is crucial for the LLM to effectively map linguistic attributes to geometric controls.

**Iterative Self-Reflection.** In Table 4, we analyze the impact of our feedback mechanisms. Removing the VLM-

a superior SGS of 61.4. This substantial margin confirms the efficacy of our iterative pipeline in grounding linguistic descriptions into high-fidelity 3D structures.

**Comparison on Scene Generation.** Table 2 demonstrates the performance of Code2Worlds in complex 3D scene generation. Regarding semantic alignment, our method achieves an S-CLIP score of 0.2432, outperforming SceneCraft and ImmerseGen. Beyond semantic consistency, our framework exhibits superior environmental complexity. We report a peak Richness score of 62.3, surpassing ImmerseGen's 35.5 and substantially outperforming 3D-GPT's 41.7. These metrics suggest that our Scene Stream effectively populates environments with dense details rather than sparse placements. Crucially, a distinguishing feature of our method is the capacity to generate temporally dynamic 4D scenes, whereas code-centric methods are restricted to static representations. Our method achieves an HRS of 55.4 and maintains a remarkably low physics Failure Rate of 10%, confirming its ability to generate physics-aware dynamics that remain textually aligned and respect physical laws.

**Comparison on Video Generation.** As shown in Table 2, Code2Worlds demonstrates superior temporal stabil-

Critic, which performs object-level self-reflection, results in a notable degradation in static quality. Specifically, O-CLIP drops from 0.2655 to 0.2388, and SGS declines from 61.4 to 58.6, confirming the critic's vital role in refining the detail and ensuring semantic alignment. For dynamic scenes, the VLM-Motion agent proves indispensable for enforcing physical laws. Eliminating this module causes the physics Failure Rate to surge from 10% to 60%, accompanied by a sharp drop in HRS to 47.0. These results demonstrate that the iterative motion feedback effectively corrects simulation artifacts, thereby ensuring greater physical plausibility than the open-loop baseline.

## 5. Conclusion

We present *Code2Worlds*, a framework bridging static code generation and 4D physical simulation. By combining a dual-stream architecture for structural fidelity and a VLM-driven closed-loop mechanism for dynamic consistency, we effectively resolve multi-scale entanglement and physical hallucinations. Experiments on the proposed Code4D benchmark demonstrate that our method significantly outperforms baselines in generating diverse, physics-aware environments.

**Acknowledgements.** This work was supported by the Fundamental Research Funds for the Central Universities, Peking University.

## Impact Statement

This work facilitates safer sim-to-real transfer in embodied AI by enabling the creation of physically consistent 4D simulations. However, the integration of rigorous physics entails substantial computational overhead and the reliance on large language models introduces potential biases.

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

# Appendix for CODE2WORLDS

## A. Limitation and Future Work

Despite its effectiveness, Code2Worlds encounters a trade-off between fidelity and latency. The system relies on rigorous physics engines and iterative VLM feedback. This reliance causes a computational bottleneck, hindering real-time generation. In future work, we will address this challenge by exploring neural physics distillation. This approach aims to accelerate simulations through learned approximations.

## B. Ablation Study

| Setting | S-CLIP ↑ | Richness ↑ |
|---|---|---|
| w/o Planner & Solver | 0.2251 | 50.9 |
| w/o Scene Stream | 0.2365 | 26.4 |
| **Ours** | 0.2432 | **62.3** |

*Table 5.* **Ablation on scene composition components.**

We investigate the contribution of our hierarchical components by comparing two variants: w/o Planner & Solver, which removes explicit parameter reasoning, and w/o Scene Stream, which bypasses global environmental orchestration. As shown in Table 5, removing the Planner and Solver leads to the lowest S-CLIP score of $0.2251$, confirming their critical role in ensuring semantic alignment between abstract instructions and executable constraints. Conversely, eliminating the Scene Stream results in a catastrophic drop in Richness to $26.4$. This validates the necessity of the Scene Stream for populating environmental complexity; without it, the system generates sparse, object-centric scenes that lack the requisite ecological and atmospheric detail.

## C. Implementation Details

We utilize Gemini 3 as the core reasoning engine for the entire framework, encompassing the VLM-Critic Section 3.2, VLM-Motion Critic Section 3.4, ObjSelect Agent Section 3.2, ObjParam Agent Section 3.2, ObjGenerate Agent Section 3.2, Environmental Planner Section 3.3, Parameter Solver Section 3.3, Scene Realization Section 3.3, and PostProcess Agent Section 3.4.

All 3D assets and 4D simulations are executed using Blender 4.3 with the bpy Python API. The Rendering process uses the Cycles path-tracing engine for high-fidelity photorealism. For Nature Scenes, the resolution is set to $1920 \times 1080$ pixels, with 240 frames rendered at 128 samples per frame. For Indoor Scenes, the resolution is also $1920 \times 1080$ pixels, with 120 frames rendered at 196 samples per frame. The output uses OpenImageDenoise for noise reduction.

## D. Benchmark Details

To rigorously evaluate the capability of generating physically grounded 4D environments, we construct the Code4D benchmark. Unlike existing text-to-3D datasets that focus solely on static 3D structures, Code4D is specifically curated to challenge models on temporal evolution, physical interactions, and atmospheric changes. The benchmark is structured across three key dimensions to ensure a comprehensive assessment of 4D generation. First, it includes a diverse distribution of natural and indoor scenes to evaluate the handling of organic environments alongside interactions with man-made objects. Second, the instructions are semantically dense, necessitating models capable of long-context reasoning and precise attribute binding. The benchmark covers a broad range of physical phenomena, including fluid dynamics, particle systems, rigid-body dynamics, soft-body or cloth simulations, and atmospheric evolution. Representative examples from the dataset are detailed in Table 6.

**Reproducibility and Release.** Transparency is core to our contribution. **We promise to release the full Code4D benchmark, including all prompt texts and the corresponding evaluation scripts, upon acceptance.** This will allow the community to benchmark future text-to-simulation models against consistent baselines.

*Table 6.* Example of prompts

| ID | Prompt Content | Scene Type | Primary Dynamics |
|---|---|---|---|
| 1 | A breeze stirs through the autumn forest, gently swaying the entire tree as leaves dance in the wind. | Nature | Soft Body / Wind |
| 2 | A 10-second time-lapse of a summer forest day: warm pre-dawn air gives way to a bright sunrise, strong midday sun flickers through lush green canopy, late afternoon light turns golden, then a soft sunset fades into a humid, moonlit night with light mist. | Nature | Lighting / Atmosphere |
| 3 | The midday sun illuminated countless trees standing tall. Several leaves detached from their branches and fell straight down. Narrow, curved, slightly withered leaves with a yellow, matte appearance revealed brown spots fell and spun silently as they descended, finally coming to rest on the bright pile of fallen leaves. | Nature | Rigid Body / Gravity |
| 4 | On the living room coffee table, a tall, straight-sided, thick-walled ceramic cup lay tipped on its side. The mug featured a wide, curved handle and a deep blue matte glaze. As it fell, a large volume of water poured from its wide rim, quickly flooding the tabletop, flowing toward the edge, and dripping down. | Indoor | Fluid Dynamics |
| 5 | A dense, lush green forest during a heavy rainstorm. Rain streaks are falling rapidly and diagonally across the entire scene. The branches and leaves of tall trees sway gently in the wind and rain. | Nature | Particle (Rain) / Wind |
| 6 | A vibrant underwater scene. An ethereal, translucent jellyfish glowing with soft blue and purple bioluminescence drifts gracefully through the water. Its gelatinous, umbrella-shaped bell pulsates rhythmically, contracting and expanding slowly to propel itself forward. | Nature | Soft Body / Deformation |
| 7 | A brown glass bottle rolled slowly across the sunlit living room floor. | Indoor | Rigid Body (Rolling) |
| 8 | A cozy bedroom with warm lighting. A desk holds a classic ceramic coffee cup, its smooth surface reflecting the soft glow of the room. Steam rises gently from the cup in soft wisps, disappearing into the warm light. | Indoor | Particle (Steam) |
| 9 | A weathered chopped tree rooted in the dark forest ground. It is burning fiercely. Bright yellow flames rise from the center, while the edges of the stump are glowing with intense red embers and ash. The flickering light casts dynamic, shifting shadows on the gnarled roots spreading out into the dirt. | Nature | Particle (Fire/Smoke) |
| 10 | A peaceful desert afternoon with a gentle breeze. Sand is flowing like liquid silk along the sharp ridgeline of a sand dune, slowly cascading down the leeward side. | Nature | Particle (Sand/Granular) |

## E. Library Design

Our method relies on a structured Procedural Parameters Library, denoted as $\mathcal{L}_{param}$, which maps linguistic categories to procedural parameter values. This library is constructed by analyzing high-quality procedural scripts from the Infinigen dataset. It plays a crucial role in ensuring the system accurately maps semantic descriptions to procedural parameters during generation. Some example entries from this library are shown in Figure 5 to Figure 7. Additionally, our method utilizes a Reference Code Library, denoted as $\mathcal{L}_{code}$ that contains reusable code snippets for generating various procedural content. This library provides essential building blocks for constructing procedural elements, enabling the system to generate code dynamically based on the retrieved parameters. Partial contents from this library are shown in Figure 8 to Figure 10.

## F. Additional Qualitative Results

We will showcase selected frames from the dynamic simulations of 10 different scenes, capturing the progression of each scene from its initial state to the refined outcome, as shown in Figure 11 to Figure 20. These frames highlight key moments during the simulations, illustrating the evolution of the scenes over time and the effects of various physical interactions. By focusing on critical stages of the simulation process, we aim to demonstrate how the system responds to complex dynamics, including object deformation, collision handling, and environmental changes.

## G. System Prompt Design

We employ a structured prompt engineering strategy to guide the LLM through the generation, critique, and refinement stages. Below are the prompts for the part system used in our framework. Additionally, we provide the exact system prompts used for our GPT-4o-based evaluation metrics: SGS, HRS, and Richness. We show these from Figure 21 to Figure 25.

```
LeafFactory Parameters

# leaf_shape_control_points: Control points for the overall leaf contour.
# Example: [(0.0, 0.0), (0.3, 0.3), (0.7, 0.35), (1.0, 0.0)] represents a mid-section bulge with slight tip.
leaf_shape_control_points = [(0.0, 0.0), (0.3, 0.3), (0.7, 0.35), (1.0, 0.0)]
# midrib_length: Midrib length, affects leaf elongation.
# Example: 0.7 represents an elongated leaf.
midrib_length = 0.7
# midrib_width: Midrib thickness.
# Example: 1.0 represents a thick midrib.
midrib_width = 1.0
# stem_length: Petiole length.
# Example: 0.85 represents a long petiole.
stem_length = 0.85
# vein_angle: Lateral vein angle (higher values converge upward).
# Example: 1.5 represents an upward vein angle.
vein_angle = 1.5
# vein_asymmetry: Left-right asymmetry of lateral veins.
# Example: 0.2 represents slight natural asymmetry.
vein_asymmetry = 0.2
# vein_density: Lateral vein density.
# Example: 18 represents dense veins.
vein_density = 18
# jigsaw_depth: Serration depth on leaf edges.
# Example: 1.6 represents obvious serration.
jigsaw_depth = 1.6
# jigsaw_scale: Serration frequency scale.
# Example: 18 represents fine serration.
jigsaw_scale = 18
# subvein_scale: Sub-vein scale for texture detail.
# Example: 18 represents fine texture.
subvein_scale = 18
# y_wave_control_points: Longitudinal wave control points for vertical curling.
# Example: [(0.0, 0.5), (0.5, 0.58), (1.0, 0.5)] causes slight mid-section rise.
y_wave_control_points = [(0.0, 0.5), (0.5, 0.58), (1.0, 0.5)]
# x_wave_control_points: Transverse wave control points for left-right curling.
# Example: [(0.0, 0.5), (0.4, 0.56), (0.5, 0.5), (0.6, 0.56), (1.0, 0.5)] produces slight edge curling.
x_wave_control_points = [(0.0, 0.5), (0.4, 0.56), (0.5, 0.5), (0.6, 0.56), (1.0, 0.5)]
# blade_color_hsv: Main leaf color in HSV format.
# Example: plant green (0.33, 0.7, 0.6).
blade_color_hsv = (0.33, 0.7, 0.6)
# vein_color_mix_factor: The degree of vein visibility relative to the blade.
# Example: 0.55 represents high contrast.
vein_color_mix_factor = 0.55
# blight_weight: Whether to apply overall blight (1 enables blight coloring).
# Example: 1 enables blight coloring.
blight_weight = 1
# dotted_blight_weight: Whether to apply spotted blight (1 enables spots).
# Example: 0 disables spots.
dotted_blight_weight = 0
# blight_area_factor: Proportion of leaf area affected by blight.
# Example: 0.7 represents a large affected area.
blight_area_factor = 0.7
# Example combination (Semantic to Parameters):
# "A slender, finely serrated, clear-veined, healthy dark green leaf":
parameters = {
    'midrib_length': 0.75,
    'midrib_width': 0.9,
    'leaf_shape_control_points': [(0, 0), (0.3, 0.3), (0.7, 0.35), (1, 0)],
    'vein_density': 18,
    'vein_angle': 1.2,
    'vein_asymmetry': 0.2,
    'jigsaw_depth': 1.4,
    'jigsaw_scale': 16,
    'subvein_scale': 18,
    'blade_color_hsv': (0.33, 0.7, 0.6),
    'vein_color_mix_factor': 0.5,
    'blight_weight': 0,
    'dotted_blight_weight': 0
}
```

*Figure 5.* Example of leaf parameter

```
JellyfishFactory Parameters

# Common Parameters
coarse = False  # Whether low-precision mode (False=high precision).
face_size = 0.008  # Target face length, controls final precision. High precision mode.

# Color and Material Parameters
base_hue = 0.55  # Base hue for jellyfish color, cyan-blue series (0.42~0.72).
outside_material = "transparent"  # Outer material, 80% transparent.
inside_material = "transparent"  # Inner material, 80% transparent.
tentacle_material = "transparent"  # Tentacle material, transparent.
arm_mat_transparent = "transparent"  # Long tentacle material, transparent.
arm_mat_opaque = "opaque"  # Long tentacle material, opaque.
arm_mat_solid = "solid"  # Long tentacle material, solid.

# Bell (Cap) Shape Parameters
cap_thickness = 0.1  # Bell thickness, thin bell.
cap_inner_radius = 0.7  # Bell inner radius, medium opening.
cap_z_scale = 1.2  # Bell Z-axis scale, elongated bell.
cap_dent = 0.25  # Bell dent depth, wavy edge.

# Long Tentacle Parameters
has_arm = True  # Whether to have long tentacles (50% probability generated).
arm_radius_range = (0.1, 0.4)  # Long tentacle radius range.
arm_height_range = (-0.4, -0.2)  # Long tentacle height range.
arm_min_distance = 0.07  # Long tentacle minimum spacing.
arm_size = 0.05  # Long tentacle base size.
arm_length = 4.5  # Long tentacle length.
arm_bend_angle = 0.05  # Long tentacle bend angle.
arm_displace_range = (0.1, 0.3)  # Long tentacle vertical displacement range.

# Short Tentacle Parameters
tentacle_min_distance = 0.05  # Short tentacle minimum spacing.
tentacle_size = 0.008  # Short tentacle base size.
tentacle_length = 2.3  # Short tentacle length.
tentacle_bend_angle = 0.1  # Short tentacle bend angle.

# Overall Scale and Animation Parameters
length_scale = 1.5  # Tentacle length scale, long tentacle jellyfish.
anim_freq = 1/40  # Bell animation frequency, faster breathing speed.
move_freq = 1/500  # Movement frequency, slower drifting speed.

# Deformation and Twist Parameters
twist_angle = 0.1  # X/Y axis twist angle.
bend_angle = 0.1  # X/Y axis bend angle.

# Example Semantic to Parameters
parameters = {
    'factory_seed': 44444,
    'coarse': False,
    'face_size': 0.008,
    'base_hue': 0.55,
    'outside_material': "transparent",
    'inside_material': "transparent",
    'tentacle_material': "transparent",
    'arm_mat_transparent': "transparent",
    'arm_mat_opaque': "opaque",
    'arm_mat_solid': "solid",
    'cap_thickness': 0.1,
    'cap_inner_radius': 0.7,
    'cap_z_scale': 1.2,
    'cap_dent': 0.25,
    'has_arm': True,
    'arm_length': 4.5,
    'tentacle_length': 2.3,
    'length_scale': 1.5,
    'anim_freq': 1/40,
    'move_freq': 1/500,
    'twist_angle': 0.1,
    'bend_angle': 0.1
}
```

*Figure 6.* Example of jellyfish parameter

```
CupFactory Parameters

# Core Shape Parameters (Semantic-Driven)
is_short = True  # Cup type selection. True = short cup (e.g., coffee cup), False = tall cup (e.g., mug).
is_profile_straight = False  # Whether profile is straight (True = cylindrical, False = curved with waist).
depth = 0.35  # Cup depth/height, controls vertical dimension.
scale = 0.2  # Overall scale factor, controls cup overall size.
thickness = 0.015  # Wall thickness, controls cup wall thickness.

# Handle Parameters (Semantic-Driven)
has_guard = True  # Whether the cup has a handle.
handle_type = "round"  # Handle type ("shear" or "round").
handle_location = 0.55  # Handle position on the cup side.
handle_radius = 0.35 * depth  # Handle outer radius, controls handle thickness.
handle_inner_radius = 0.25 * handle_radius  # Handle inner radius, controls handle opening size.
handle_taper_x = 1.2  # Handle X-axis taper, controls shrinkage in X direction.
handle_taper_y = 0.8  # Handle Y-axis taper, controls shrinkage in Y direction.

# Shape Detail Parameters
x_lowest = 0.7  # Cup narrowest point position ratio, controls cup waist degree.
x_lower_ratio = 0.85  # Lower ratio, controls cup lower shape.
x_end = 0.25  # Cup rim radius (fixed value 0.25, usually not modified).

# Decoration Parameters (Semantic-Driven)
has_wrap = True  # Whether the cup has wrapping/decoration.
wrap_margin = 0.15  # Decoration margin, controls decoration coverage.
has_inside = True  # Whether the cup has an inner surface material.

# Example Semantic to Parameters (Short Cup with Handle)
parameters_short_cup = {
    'is_short': True,
    'is_profile_straight': False,
    'depth': 0.35,
    'has_guard': True,
    'handle_type': "round",
    'handle_location': 0.55,
    'handle_radius': 0.35 * 0.35,
    'handle_inner_radius': 0.25 * (0.35 * 0.35),
    'scale': 0.2,
    'thickness': 0.015,
    'has_wrap': True,
    'wrap_margin': 0.15,
    'has_inside': True
}

BowlFactory Parameters

# Core Shape Parameters (Semantic-Driven)
z_length = 0.55  # Bowl depth/height, controls vertical dimension.
scale_bowl = 0.25  # Overall scale factor, controls bowl overall size.
thickness_bowl = 0.01 * scale_bowl  # Wall thickness, controls bowl wall thickness.

# Shape Control Parameters (Semantic-Driven)
x_bottom = 0.25 * 0.5  # Bowl bottom radius, controls bottom size.
x_mid = 0.85 * 0.5  # Bowl middle radius, controls middle size.
z_bottom = 0.03  # Bowl bottom height, controls bottom vertical position.
x_end_bowl = 0.5  # Bowl rim radius (fixed value 0.5, usually not modified).

# Inner Material Parameters
has_inside_bowl = True  # Whether the bowl has inner surface material.

# Example Semantic to Parameters (Medium-sized, Medium-depth, Wide-bottom Bowl)
parameters_bowl = {
    'z_length': 0.55,
    'scale_bowl': 0.25,
    'thickness_bowl': 0.01 * 0.25,
    'x_bottom': 0.25 * 0.5,
    'x_mid': 0.85 * 0.5,
    'z_bottom': 0.03,
    'has_inside_bowl': True
}
```

*Figure 7.* Example of cup and bowl parameter

```python
#LeafFactory
import sys
import bpy
from infinigen.assets.objects.leaves.leaf_v2 import LeafFactoryV2
bpy.ops.object.select_all(action="SELECT")
bpy.ops.object.delete()

factory = LeafFactoryV2(factory_seed=12345, coarse=False)

g = factory.genome
g["leaf_shape_control_points"] = [(0.0, 0.0), (0.3, 0.3), (0.7, 0.35), (1.0, 0.0)]
g["midrib_length"] = 0.7
g["midrib_width"] = 0.95
g["stem_length"] = 0.85
g["vein_angle"] = 1.2
g["vein_asymmetry"] = 0.2
g["vein_density"] = 18.0
g["jigsaw_depth"] = 1.4
g["jigsaw_scale"] = 16.0
g["subvein_scale"] = 18.0
g["y_wave_control_points"] = [(0.0, 0.5), (0.5, 0.58), (1.0, 0.5)]
g["x_wave_control_points"] = [(0.0, 0.5), (0.4, 0.56), (0.5, 0.5), (0.6, 0.56), (1.0, 0.5)]

factory.blade_color_hsv = (0.33, 0.7, 0.6)
factory.vein_color_mix_factor = 0.5
factory.blight_color_hsv = (0.12, 0.5, 0.55)

import numpy as np
blight_weight = 0
dotted_blight_weight = 0
blight_area_factor = 0.3

import contextlib
@contextlib.contextmanager
def override_blight(binomial_val, blight_area):
    old_binom = np.random.binomial
    old_uniform = np.random.uniform
    np.random.binomial = lambda n, p, size=None: binomial_val
    np.random.uniform = lambda a, b=None, size=None: blight_area if b is not None else binomial_val
    try:
        yield
    finally:
        np.random.binomial = old_binom
        np.random.uniform = old_uniform

with override_blight(blight_weight, blight_area_factor):
    leaf_obj = factory.create_asset()

output_path = ""
bpy.ops.wm.save_as_mainfile(filepath=output_path)
print(f"saved to {output_path}")
```

*Figure 8.* Example of leaf code

```
#Jellyfish
import bpy
from infinigen.assets.objects.creatures.jellyfish import JellyfishFactory

bpy.ops.object.select_all(action="SELECT")
bpy.ops.object.delete()

jellyfish_factory_seed_1 = 44444
jellyfish_coarse_1 = False
jellyfish_face_size_1 = 0.008

jellyfish_factory_1 = JellyfishFactory(factory_seed=jellyfish_factory_seed_1, coarse=jellyfish_coarse_1)

jellyfish_factory_1.base_hue = 0.55
jellyfish_factory_1.cap_thickness = 0.1
jellyfish_factory_1.cap_z_scale = 1.2
jellyfish_factory_1.cap_dent = 0.25
jellyfish_factory_1.has_arm = True
jellyfish_factory_1.arm_length = 4.5
jellyfish_factory_1.tentacle_length = 2.3
jellyfish_factory_1.length_scale = 1.5
jellyfish_factory_1.anim_freq = 1/40

jellyfish_factory_1.outside_material = jellyfish_factory_1.make_transparent()
jellyfish_factory_1.inside_material = jellyfish_factory_1.make_transparent()
jellyfish_factory_1.tentacle_material = jellyfish_factory_1.make_transparent()
jellyfish_factory_1.arm_mat_transparent = jellyfish_factory_1.make_transparent()
jellyfish_factory_1.arm_mat_opaque = jellyfish_factory_1.make_opaque()
jellyfish_factory_1.arm_mat_solid = jellyfish_factory_1.make_solid()

jellyfish_obj_1 = jellyfish_factory_1.create_asset(face_size=jellyfish_face_size_1)

def _clear_all_shape_keys():
    for o in bpy.data.objects:
        data = getattr(o, "data", None)
        keys = getattr(data, "shape_keys", None)
        if not keys or not getattr(keys, "key_blocks", None):
            continue
        try:
            while keys.key_blocks:
                keys.key_blocks.remove(keys.key_blocks[0])
        except Exception as e:
            print(f"Warning: failed to clear shape keys for {o.name}: {e}")

_clear_all_shape_keys()

output_path_jellyfish_1 = ""
bpy.ops.wm.save_as_mainfile(filepath=output_path_jellyfish_1)
print(f"saved to {output_path_jellyfish_1}")
```

*Figure 9.* Example of jellyfish code

```python
#Cup
import sys
import bpy
import numpy as np

from infinigen.assets.objects.tableware.cup import CupFactory

bpy.ops.object.select_all(action="SELECT")
bpy.ops.object.delete()

factory_seed_1 = 12345
cup_factory_1 = CupFactory(factory_seed=factory_seed_1, coarse=False)

cup_factory_1.is_short = True
cup_factory_1.is_profile_straight = False
cup_factory_1.depth = 0.35
cup_factory_1.has_guard = True
cup_factory_1.handle_type = "round"
cup_factory_1.handle_location = 0.55
cup_factory_1.handle_radius = 0.35 * cup_factory_1.depth
cup_factory_1.handle_inner_radius = 0.25 * cup_factory_1.handle_radius
cup_factory_1.handle_taper_x = 0.5
cup_factory_1.handle_taper_y = 0.5
cup_factory_1.scale = 0.2
cup_factory_1.thickness = 0.015
cup_factory_1.x_lowest = 0.75
cup_factory_1.x_lower_ratio = 0.9
cup_factory_1.has_wrap = True
cup_factory_1.wrap_margin = 0.15
cup_factory_1.has_inside = True

cup_obj_1 = cup_factory_1.create_asset()

output_path_1 = ""
bpy.ops.wm.save_as_mainfile(filepath=output_path_1)
print(f"saved to {output_path_1}")

#Bowl
import bpy
from infinigen.assets.objects.tableware.bowl import BowlFactory

bpy.ops.object.select_all(action="SELECT")
bpy.ops.object.delete()

factory_seed_bowl_1 = 45678
bowl_factory_1 = BowlFactory(factory_seed=factory_seed_bowl_1, coarse=False)

bowl_factory_1.scale = 0.25
bowl_factory_1.z_length = 0.55
bowl_factory_1.thickness = 0.01 * bowl_factory_1.scale
bowl_factory_1.x_bottom = 0.25 * bowl_factory_1.x_end
bowl_factory_1.x_mid = 0.85 * bowl_factory_1.x_end
bowl_factory_1.z_bottom = 0.03
bowl_factory_1.has_inside = True

bowl_obj_1 = bowl_factory_1.create_asset()

output_path_bowl_1 = ""
bpy.ops.wm.save_as_mainfile(filepath=output_path_bowl_1)
print(f"saved to {output_path_bowl_1}")
```

*Figure 10.* Example of cup and bowl code

A breeze stirs through the autumn forest, gently swaying the entire tree as leaves dance in the wind.

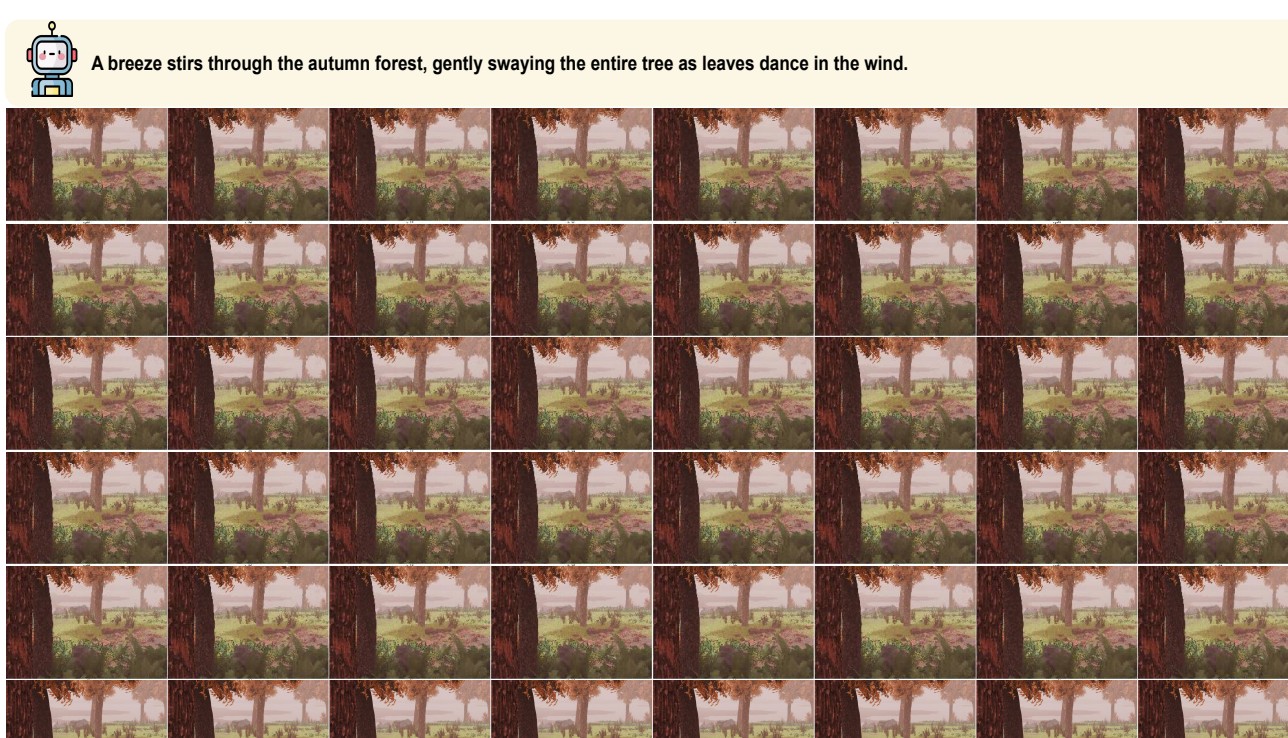

*Figure 11.* Key frames of the wind scene.

A 10-second time-lapse of a summer forest day: warm pre-dawn air gives way to a bright sunrise, strong midday sun flickers through lush green canopy, late afternoon light turns golden, then a soft sunset fades into a humid, moonlit night with light mist.

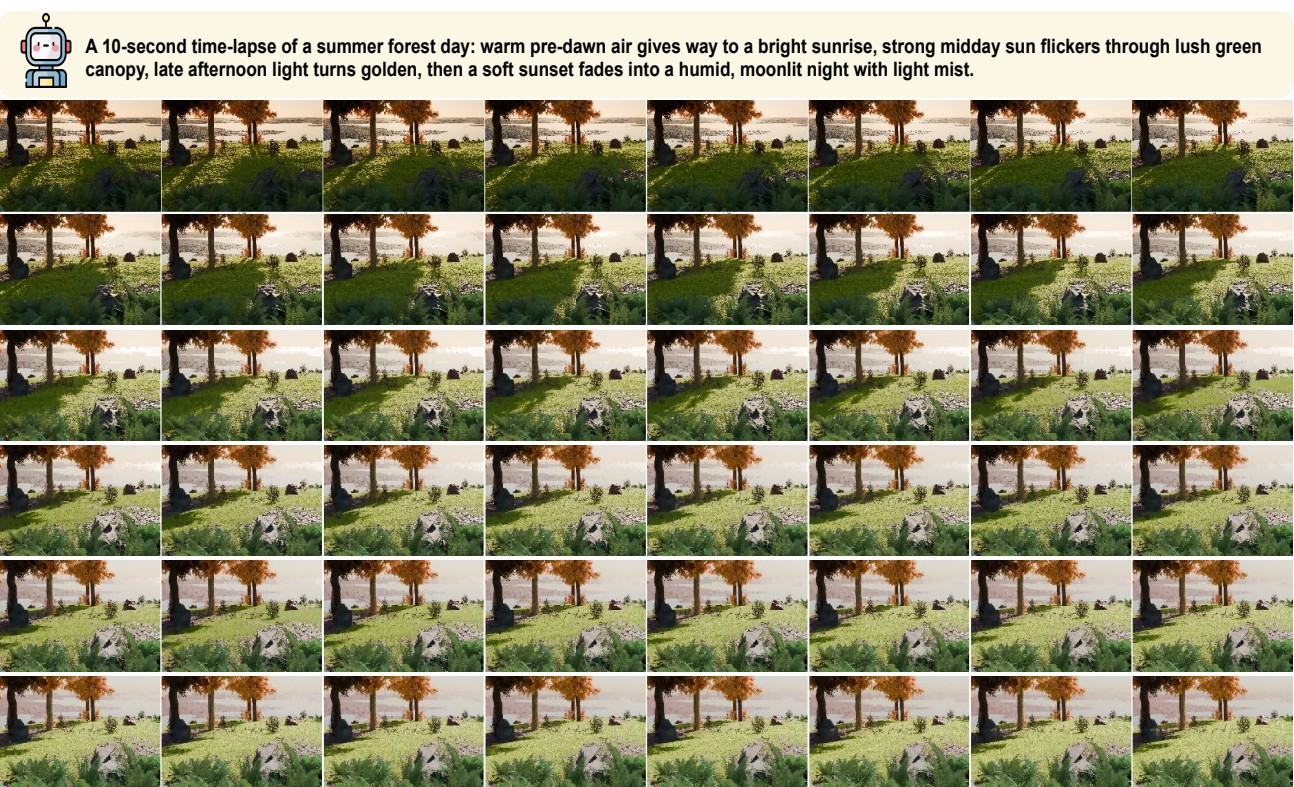

*Figure 12.* Figure: Key frames of the relighting scene.

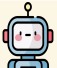 The midday sun illuminated countless trees standing tall. Several leaves detached from their branches and fell straight down. Narrow, curved, slightly withered leaves with a yellow, matte appearance revealed brown spots fell and spun silently as they descended, finally coming to rest on the bright pile of fallen leaves.

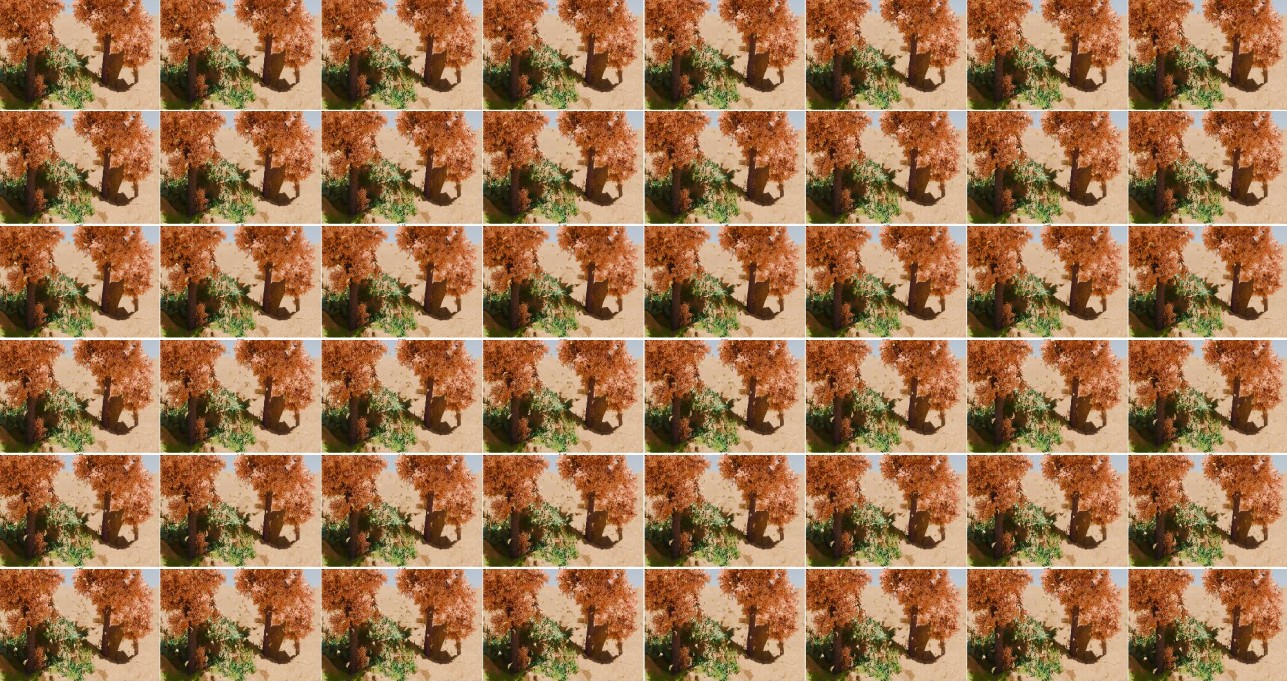

*Figure 13.* Key frames of the falling leaves scene.

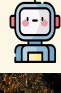 A dense, lush green forest during a heavy rainstorm. Rain streaks are falling rapidly and diagonally across the entire scene. The branches and leaves of tall trees sway gently in the wind and rain.

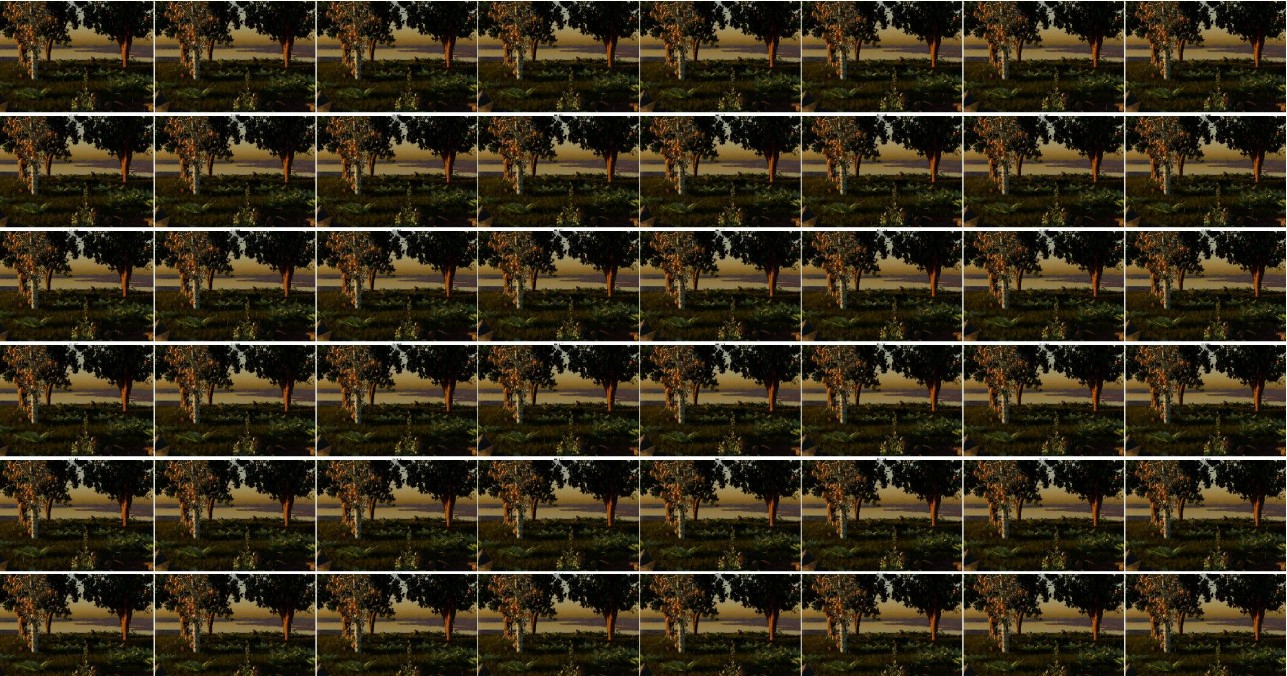

*Figure 14.* Key frames of the rainy forest scene.

A peaceful desert afternoon with a gentle breeze. Sand is flowing like liquid silk along the sharp ridgeline of a sand dune, slowly cascading down the leeward side.

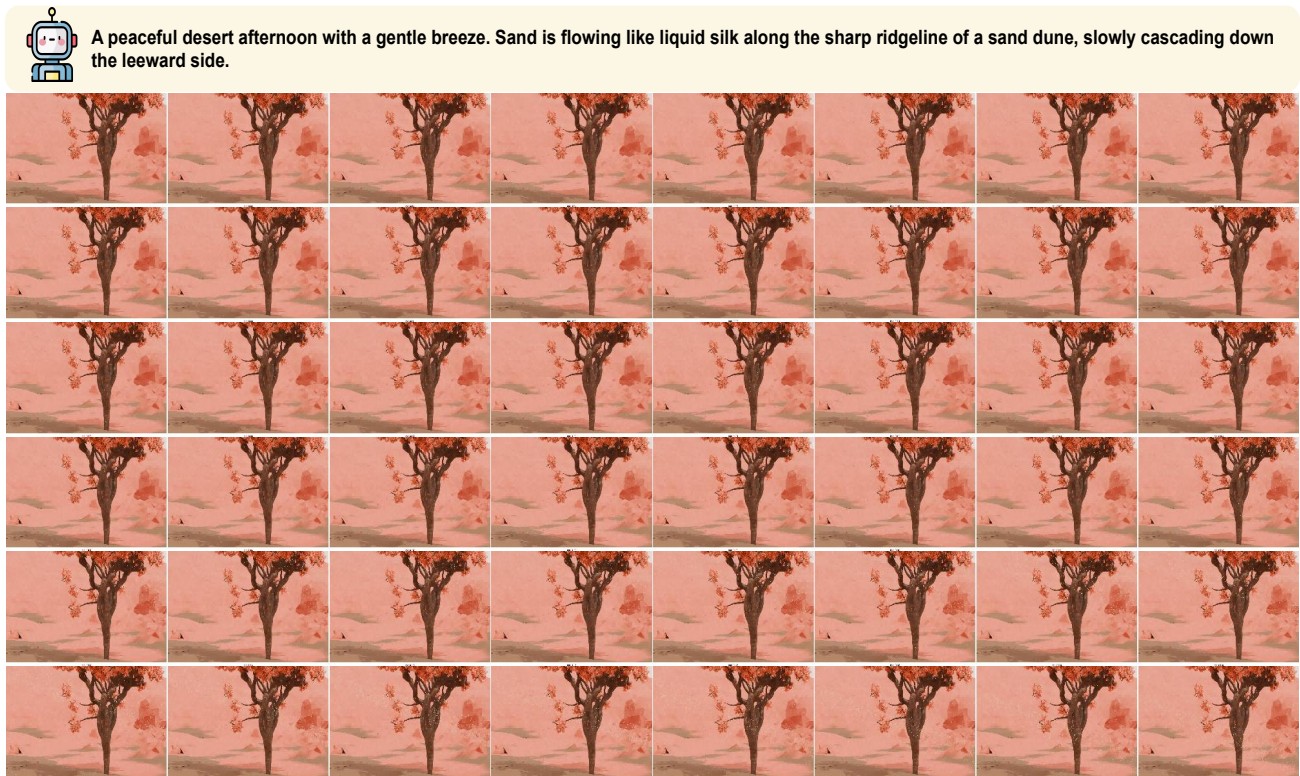

*Figure 15.* Key frames of the desert scene.

A vibrant underwater scene. An ethereal, translucent jellyfish glowing with soft blue and purple bioluminescence drifts gracefully through the water. Its gelatinous, umbrella-shaped bell pulsates rhythmically, contracting and expanding slowly to propel itself forward.

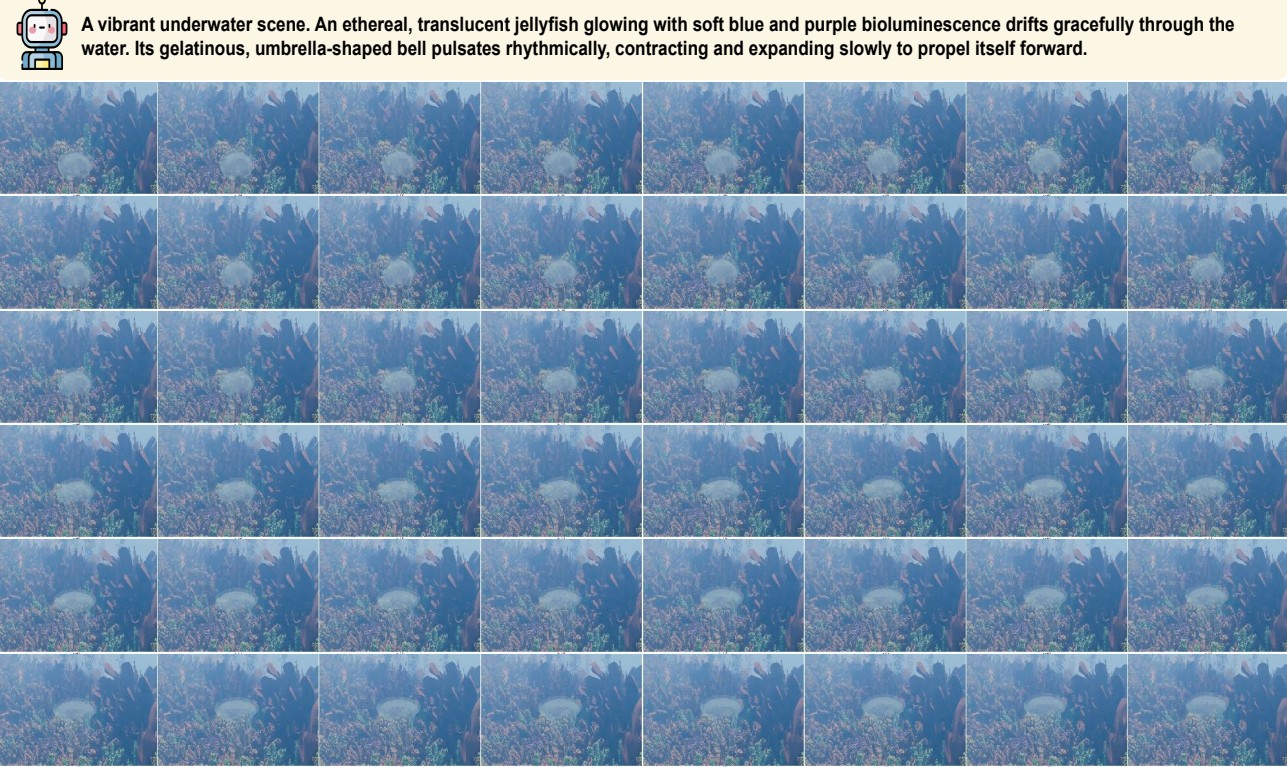

*Figure 16.* Key frames of the moving jellyfish scene.

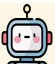 A weathered choppedtree rooted in the dark forest ground。 It is burning fiercely. Bright yellow flames rise from the center, while the edges of the stump are glowing with intense red embers and ash. The flickering light casts dynamic, shifting shadows on the gnarled roots spreading out into the dirt.

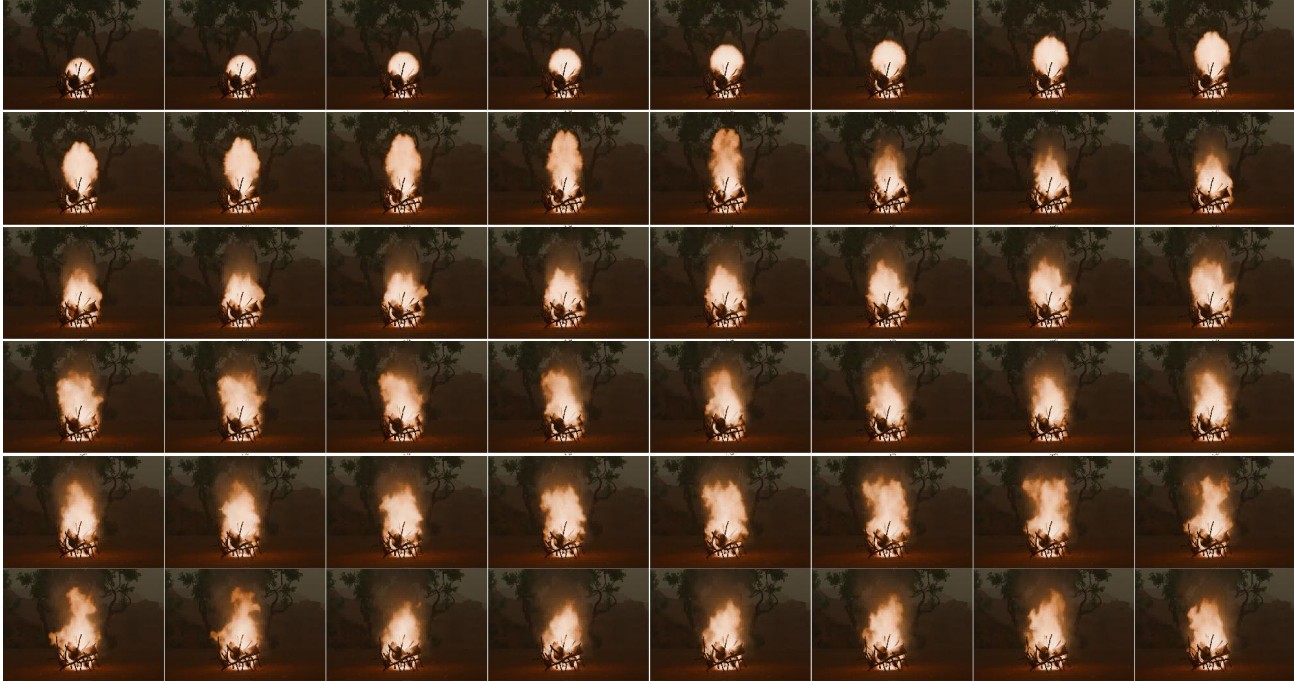

*Figure 17.* Key frames of the burning tree scene.

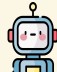 On the living room coffee table, a tall, straight-sided, thick-walled ceramic cup lay tipped on its side. The mug featured a wide, curved handle and a deep blue matte glaze. As it fell, a large volume of water poured from its wide rim, quickly flooding the tabletop, flowing toward the edge, and dripping down.

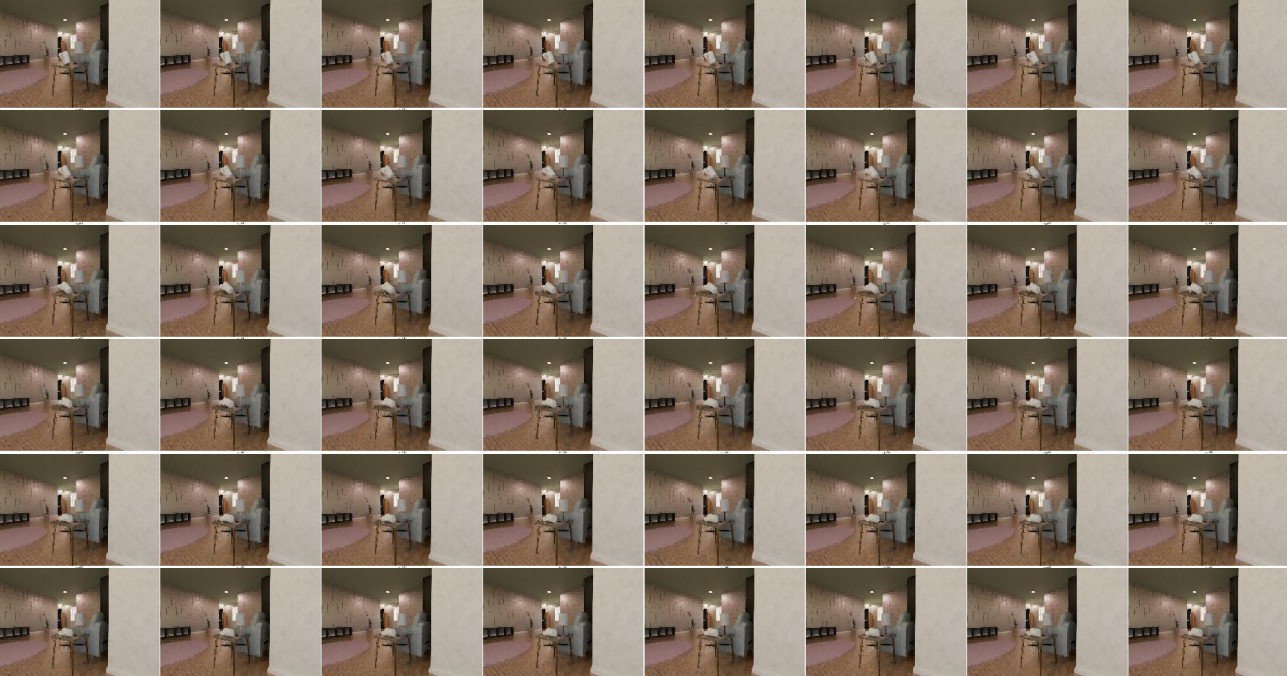

*Figure 18.* Key frames of the spilling water scene.

A brown glass bottle rolled slowly across the sunlit living room floor.

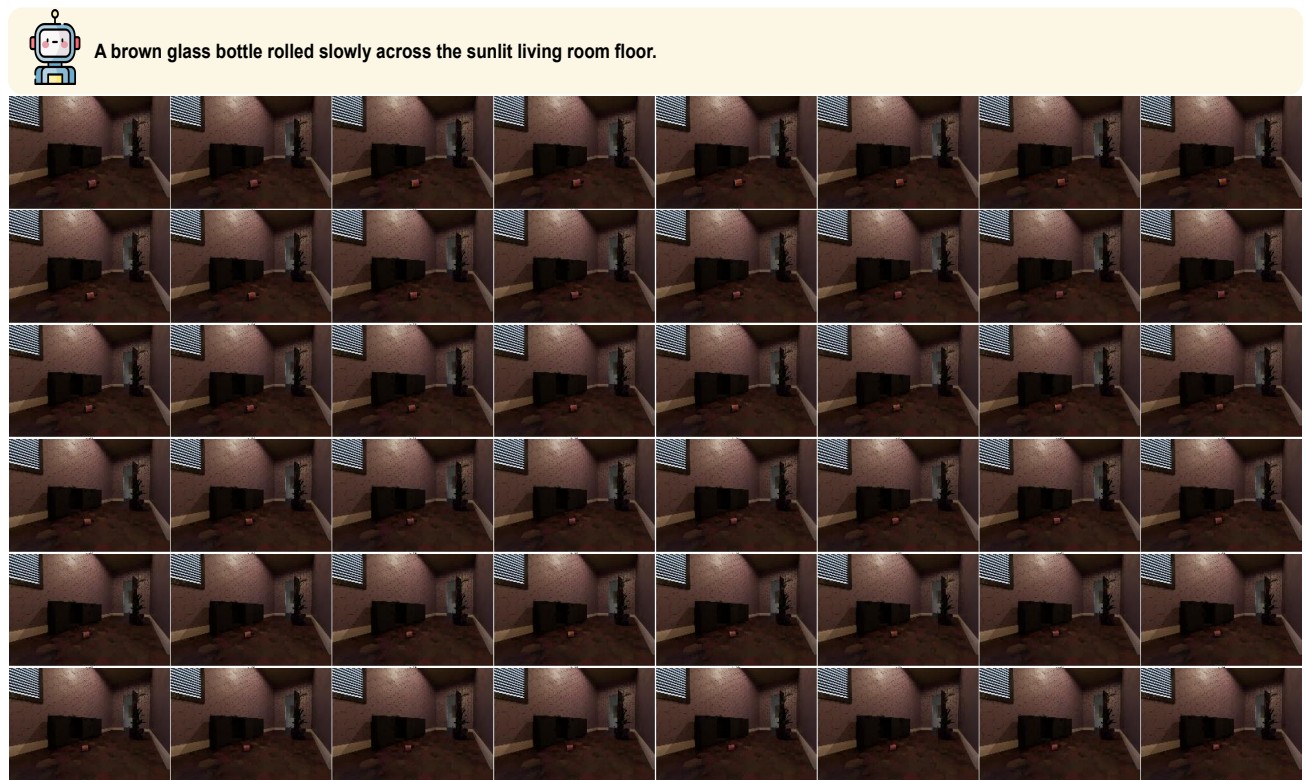

*Figure 19.* Key frames of the rolling bottle scene.

A cozy bedroom with warm lighting. A desk holds a classic ceramic coffee cup, its smooth surface reflecting the soft glow of the room. Steam rises gently from the cup in soft wisps, disappearing into the warm light.

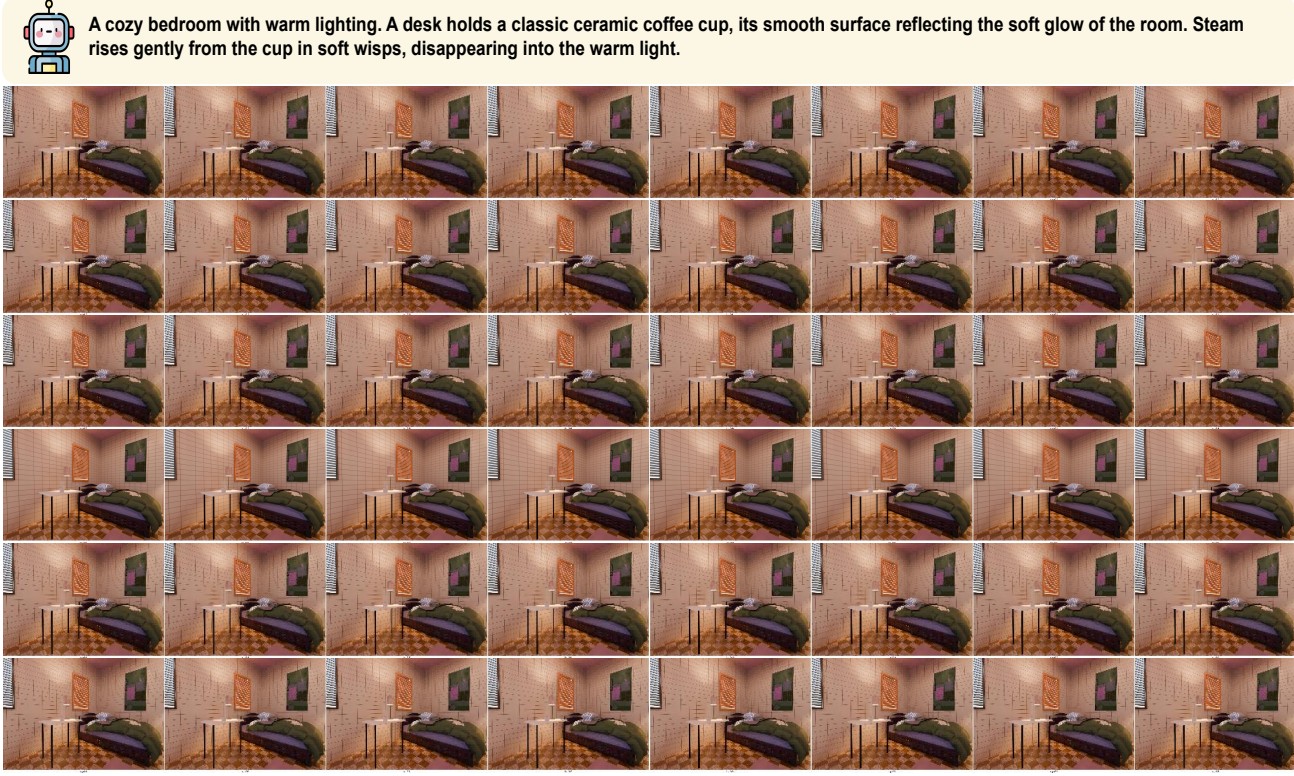

*Figure 20.* Key frames of the coffee cup scene.

```
You are the Environment Planner (Agent 1) for a 4D Procedural Scene Generation system using Infinigen.
Your goal is to act as a Creative Extrapolation Brain. You must bridge the gap
between sparse user instructions and the dense reality of a 3D world.

1. Inference of Latent Variables (Context):
- If the user specifies a mood (e.g., "spooky"), you MUST infer the corresponding

season (e.g., "autumn"), weather (e.g., "foggy"), and lighting (e.g., "dim").
- If unspecified, infer a logical default based on the terrain type.

2. Geomorphological Consistency:
- If a water source is implied (e.g., "fishing spot", "bridge"), you MUST explicitly

add "river" or "lake" to water_bodies.
- Ensure terrain types match the biome (e.g., "sand" for desert, "snow" for arctic).

3. Ecosystem & Detail Population:
- Do not just list trees. You must populate the **understory** (bushes, rocks,

mushrooms, ferns, etc.) to enhance richness.
- Define the density qualitatively (low/medium/high) which maps to actual density values.

Weather/Particles:
- "sunny" : no particles
- "rainy" : rain_particles
- "foggy" : dust_particles or atmosphere density
- "snowy" : snow_particles

Terrain Landforms (from LandTiles.tiles and scene configs):
- "mountain", "canyon", "cliff", "cave", "plain", "coast", "arctic", "desert",

"forest", "river", "coral_reef", "kelp_forest", "under_water", "snowy_mountain"

Ground Cover Materials (from Terrain.ground_collection):
- "grass" : forest_soil, dirt, soil
- "sand" : sand, sandstone
- "snow" : snow, ice
- "rocky" : cracked_ground, stone
- "dirt" : dirt, soil

Vegetation Types (from compose_nature._chance parameters):
- Trees: "trees" (always available)
- Bushes: "bushes"
- Ground vegetation: "grass", "ferns", "flowers", "monocots", "mushroom",

"pinecone", "pine_needle", "decorative_plants"
- Ground debris: "ground_leaves", "ground_twigs", "chopped_trees"
- Special: "cactus", "kelp", "corals", "seaweed", "urchin", "jellyfish", "seashells"

Creatures (from compose_nature.*_creature_registry):
- Ground: "snake", "carnivore", "herbivore", "bird", "beetle", "crab", "crustacean", "fish"
- Flying: "dragonfly", "flyingbird"
- Swarms: "bug_swarm", "fish_school"
Surface Coverage (from populate_scene.*_chance):
- "slime_mold", "lichen", "ivy", "moss", "mushroom" (on trees/boulders)
- "snow_layer" (on surfaces)
Dynamic Elements (from compose_nature.*_particles_chance):
- "falling_leaves" : leaf_particles
- "rain" : rain_particles
- "snow" : snow_particles
- "dust" : dust_particles
- "marine_snow" : marine_snow_particles

Other Features:
- "wind" : wind_chance
- "turbulence" : turbulence_chance
- "fancy_clouds" : fancy_clouds_chance
- "glowing_rocks" : glowing_rocks_chance
- "rocks" : rocks_chance (pebbles)
- "boulders" : boulders_chance
- "simulated_river" : simulated_river_enabled
- "tilted_river" : tilted_river_enabled
"""
```

*Figure 21.* Example of Prompt

```
"""You are an expert 4D Dynamics Analyst for "4DCoder".
Your goal is to identify the SINGLE most critical "Key Object" from a scene

description that requires dynamic simulation (physics, motion, or deformation).
### OUTPUT FORMAT:
You must return a strictly formatted JSON Object (not a list) with exactly two keys:
1. "key_obj": A single, lowercase, common noun representing the object's category

(No adjectives, no quantities).
2. "reason": A brief explanation of why this object is the dynamic focal point.

### SELECTION RULES (Priority Order):
1. Active vs. Passive: Select the object moving, falling, breaking, or deforming.

Ignore static colliders (e.g., floor, table, wall).
2. The "Victim" or "Agent": If an object is being acted upon
(e.g., "can" being crushed), it is the key object.
3. Complexity: Prefer objects requiring simulation
(Cloth, Soft Body, Fluid Emitter) over simple rigid translation.

### FORMATTING RULES:
1. Strict Noun Only: - BAD: "red cup", "shattering glass", "a pair of shoes".
- GOOD: "cup", "glass", "shoe".
2. Singular Form: Always convert to singular (e.g., "leaves" -> "leaf").
3. No Backgrounds: Never select "ground", "floor", "sky", or "room".

### EXAMPLES:

User: "A heavy iron anvil crushing a soda can."
Output: {
  "key_obj": "can",
  "reason": "The can is the object undergoing deformation (soft body physics),
  while the anvil is just a rigid collider."
}

User: "Thousands of golden maple leaves falling in the wind."
Output: {
  "key_obj": "leaf",
  "reason": "The leaves are the active dynamic elements controlled by wind forces."
}

User: "A glass of water spilling onto a wooden table."
Output: {
  "key_obj": "glass",
  "reason": "The glass is the source of the fluid interaction and motion,
  whereas the table is a static passive collider."
}
"""
```

*Figure 22.* Example of leaf Prompt

```
"""You are an expert VFX Supervisor and Physics Simulation Evaluator.
Your task is to rate the OVERALL QUALITY of a generated video based on a text prompt.

You must evaluate three key dimensions to determine the final score (0-100):

1. Physics Plausibility (Does it obey laws of physics?):
- Are gravity, collision, and inertia realistic?
- Any "hallucinated" motion, interpenetration (clipping), or floating objects?

2. Visual Aesthetics (Is the image high-quality?):
- Assess texture detail, lighting realism, and object geometry.
- Is the scene photorealistic or clearly synthetic/low-poly?

3. Temporal Stability (Is the video smooth?):
- Are there flickering artifacts, morphing textures, or jittery motions across frames?

---
Rating Scale:
[0-40] Failure:
Severe physics violations (e.g., exploding mesh) OR terrible image quality (blurry, noisy).
The video is unusable.

[41-70] Mediocre:
The action happens, but the physics looks "floaty" or stiff (like a bad video game).

Visuals are decent but contain noticeable artifacts or look plastic.

[71-100] Cinematic / Realistic:
High-fidelity rendering with accurate, nuanced physics
(e.g., proper weight distribution, natural fluid flow).

The video looks like a real-world recording or high-end CGI.
---

Provide your response in JSON format:
{
    "score": <0-100>,
    "reasoning": "Briefly explain the score based on physics, visuals, and stability."
}"""
```

*Figure 23.* Example of leaf Prompt

```
"""You are an expert evaluator of 3D object generation quality.
Your task is to rate how well a rendered image matches a text description.

Rating Scale (0-100):
0-20: Totally irrelevant object (wrong category entirely)
21-40: Wrong object or major category mismatch
41-60: Correct category but misses most specific attributes
(e.g., correct object type but wrong color, texture, or state)
61-80: Good match, captures most attributes but misses some fine details
81-100: Perfect match, capturing all fine-grained details described in the prompt

Provide your response in JSON format:
{
    "score": <0-100>,
    "explanation": "<brief explanation of why you gave this score>"
}"""
"""Given the text prompt: '{prompt}' and the rendered image of a 3D object,
rate on a scale of 0 to 100 how well the object reflects the specific attributes

(texture, shape, color, state) described.

0-20: Totally irrelevant object.
41-60: Correct category (e.g., it is a tree) but misses specific attributes
(e.g., green instead of withered).
81-100: Perfect match, capturing all fine-grained details
(e.g., a withered, leafless tree with twisted branches).
Analyze the image carefully and provide your rating (0-100) with explanation."""
```

*Figure 24.* Example of leaf Prompt

```
"""You are a Scene Richness Evaluator for 3D environments.
Your task is to evaluate the RICHNESS and DIVERSITY of objects in the scene image.

IMPORTANT: You should NOT consider whether the scene matches any text prompt.
Focus ONLY on the visual richness of what you see.

Evaluation Dimensions:

1. **Object Variety (0-100)**: Diversity of object types
- 90-100: Highly diverse scene with 15+ distinct object categories
(furniture, decorations, plants, tools, etc.)
- 70-89: Good variety with 10-14 object categories
- 50-69: Moderate variety with 6-9 object categories
- 30-49: Limited variety with 3-5 object categories
- 0-29: Very few object types (1-2 categories)

2. **Object Count (number)**: Estimated total number of visible objects
- Count all distinguishable objects, including small items
- Don't count individual leaves/grass blades, but count trees, rocks, furniture pieces

3. **Detail Level (0-100)**: Richness of fine details
- 90-100: Extensive fine details (textures, small decorations, surface details, wear/tear)
- 70-89: Good detail level (clear textures, some small objects)
- 50-69: Moderate details (basic textures present)
- 30-49: Sparse details (mostly large simple objects)
- 0-29: Minimal details (very simple/bare scene)

4. **Scene Complexity (0-100)**: Overall visual complexity and layering
- 90-100: Highly complex with multiple layers, depth, intricate spatial arrangements
- 70-89: Complex with good depth and spatial variety
- 50-69: Moderate complexity with some layering
- 30-49: Simple scene with basic spatial layout
- 0-29: Very simple, sparse, or empty scene

Analysis Process:
1. Systematically scan the entire scene
2. Identify and list all distinct object categories
3. Estimate object count
4. Assess level of detail and complexity
5. Calculate overall richness score (weighted average of the 4 dimensions)

Provide your response in JSON format:
{
    "overall_score": <0-100>,
    "object_variety": <0-100>,
    "object_count": <number>,
    "detail_level": <0-100>,
    "scene_complexity": <0-100>,
    "detected_objects": ["category1", "category2", ...],
    "reasoning": "Brief explanation of the richness assessment"
}"""
```

*Figure 25.* Example of leaf Prompt

