# OpenReview forum: "Code2Worlds: Empowering Coding LLMs for 4D World Generation"
_ICML.cc/2026/Conference — ICML 2026 regular_

### Official Review · Reviewer_g5tr · 2026-03-09

**Soundness:** 2
**Presentation:** 2
**Significance:** 2
**Originality:** 3
**Overall Recommendation:** 4
**Confidence:** 3

**Summary:**

This paper presents Code2Worlds, a framework that uses coding LLMs to generate physically grounded 4D scenes (3D + dynamics) from text descriptions. The core idea is to formulate 4D generation as language-to-simulation code generation targeting Blender+Infinigen. The system has two main design choices: a dual-stream architecture that separates object generation (via retrieval-augmented parametric generation from Infinigen's procedural library) from scene-level environmental orchestration, and a closed-loop refinement mechanism where a VLM critic evaluates rendered outputs (both static snapshots and video rollouts) and feeds back corrections. The authors also introduce Code4D, a small benchmark of 10 prompts spanning nature and indoor scenes with various dynamics (wind, fluid, fire, rigid body, etc.). Results show improvements over prior code-to-scene methods (41% SGS gain, 49% Richness gain) and competitive temporal stability against video diffusion models.

**Compliance With Llm Reviewing Policy:**

Affirmed.

**Final Justification:**

The paper is original and technically interesting, and my main initial concerns were about missing evaluation and experimental details rather than the core idea. The rebuttal addressed these points sufficiently, which increased my confidence in the work and led me to raise my score.

**Key Questions For Authors:**

1. How does the system handle objects not in Infinigen's factory catalog? If so, can it do anything at all? This seems like a fundamental limitation that deserves honest discussion.
2. Have you validated the GPT-4o evaluation metrics (SGS, HRS, Richness) against human judgments? Even a small-scale correlation study would help.
3. What's the end-to-end generation time per scene, including all VLM feedback iterations and Blender rendering?

**Limitations:**

The paper briefly mentions computational overhead and LLM biases in the impact statement, and latency in Appendix A. But it doesn't address the most obvious limitation: the system only works for object categories that Infinigen supports. This is a major scope restriction that should be clearly stated upfront.

**Strengths And Weaknesses:**

**Strengths:**
- The problem is well-motivated, extending procedural code generation from static 3D to physically grounded 4D is a natural and important next step, and the two identified challenges (multi-scale entanglement, semantic-physical gap) are real.
- The dual-stream separation of object detail from scene layout is pragmatic and well-validated. The ablation showing SGS dropping from 61.4 to 23.5 without retrieval is convincing.
- The VLM-Motion closed-loop for dynamics is the most interesting contribution. Rendering a rollout, having a VLM judge the physics, and iteratively refining simulation code is novel here. Failure rate jumping from 10% to 60% without it backs this up.
- The qualitative results (fire, water spill, jellyfish, relighting) are genuinely impressive and look like plausible Blender renders with coherent dynamics.

**Weaknesses:**
- The benchmark size is never stated. Table 6 shows 10 "example" prompts but the paper doesn't report how many total prompts Code4D contains. If benchmark is just 10 prompts, that's more of a demo, not a benchmark. A 10% failure rate on 10 samples means exactly one failure. No variance estimates, no statistical reliability.
- Evaluation leans heavily on GPT-4o as judge (it won't cause an issue but also why not a more recent version?) with no human evaluation and no validation that these metrics correlate with human perception in this domain.
- The system only works for objects in Infinigen's factory catalog (leaves, cups, jellyfish, etc.). This is a fundamental generality limitation the paper doesn't adequately discuss.
- The comparison with video diffusion models on VBench metrics is somewhat unfair, deterministic Blender renders will naturally beat stochastic diffusion samples on temporal flickering and smoothness. The interesting comparison would be on semantic/physical quality.
- No reporting of generation time or cost, despite iterative VLM loops and Blender rendering being presumably expensive.

---

> ### Author Rebuttal · Authors · 2026-03-31
>
> ## Response to Reviewer g5tr
>
> We thank the reviewer for the thoughtful and constructive feedback.
>
> ### Regarding W1: benchmark
> We thank the reviewer for this comment. To our knowledge, Code4D is the first benchmark for evaluating physically grounded 4D world generation from text. It currently contains approximately 50 prompts, with the 10 shown in Table 6 serving as representative examples. Our design prioritizes systematic coverage over raw volume, spanning diverse dynamics categories (rigid-body, particle, fluid, fire/smoke, and environmental effects) and varying complexity levels. Moreover, each sample is a full end-to-end test involving program generation, physics simulation, and rendering, making construction and evaluation substantially more costly than standard static text-to-3D datasets. We are also continuing to expand Code4D to a substantially larger scale.
>
> ### Regarding W2 and Q2: human evaluation and validation of GPT-based metrics
> We thank the reviewer for raising this important concern. We refer the reviewer to our response to **Reviewer e4Ub, Regarding W1**, for the full human-evaluation protocol, correlation statistics, and detailed results. Briefly, to address this issue, we conducted an additional human evaluation on SGS, Richness, and HRS, where annotators rated generated results on a 1–5 scale. The human results are strongly consistent with the GPT-based metrics, both in overall ranking and in correlation analysis, and support the same conclusion: Code2Worlds ranks best on semantic fidelity and scene richness, while also showing better physical plausibility than the ablated variant.
>
> ### Regarding W3 and Q1: Infinigen vocabulary limitation and unsupported objects
> We thank the reviewer for raising this important question. We refer the reviewer to our response to **Reviewer e4Ub, Regarding W3 and Q2**, for more details. Briefly, our framework is a general language-to-simulation pipeline, not a method fundamentally restricted to Infinigen. We use Infinigen in the current experiments because it offers a convenient procedural interface and executable physical controls, but the same pipeline can be extended to other asset sources through a unified downstream object interface. To improve vocabulary coverage, our framework adopts a three-stage graceful-degradation mechanism: local knowledge base retrieval, external 3D asset retrieval, and direct LLM generation when retrieval is unavailable. Since all three stages produce a unified object representation, the downstream PostProcess Stream and physics-aware closed-loop refinement can be applied consistently regardless of object source.
>
> ### Regarding W4: fairness of comparison
> | Method | Human Semantic Alignment  | Failure Rate  |
> |--------|--------------------------:|---------------:|
> | Stable Video Diffusion | 3.5 | 50% |
> | AnimateDiff | 3.2 | 70% |
> | CogVideoX | 3.7 | 50% |
> | Hunyuan | 4.0 | 30% |
> | Code2Worlds | 4.1 | 10% |
>
> We thank the reviewer for this important point. We agree that VBench temporal metrics alone are insufficient to claim semantic or physical superiority over video diffusion models, since deterministic Blender rendering can naturally improve flicker and smoothness. In our paper, VBench was intended only as a temporal-stability measure, while Failure Rate captures objective physical violations. To address this fairness concern directly, we also conducted a human semantic evaluation, in which annotators rated semantic alignment on a 1–5 scale, and we report these ratings alongside the Failure Rate on the same video baselines. Code2Worlds achieves the best semantic score and the lowest physical failure rate, demonstrating that its advantage extends beyond deterministic temporal smoothness to semantic alignment and physical correctness. We will revise the paper to clarify this interpretation and add the above comparison of human semantic and physical failures as a fairer evaluation.
>
> ### Regarding W5 and Q3: generation time and cost
> We thank the reviewer for pointing this out. To better address the practical applicability of our framework, we now report the end-to-end runtime and cost statistics. In this setup (single RTX 4090, 120 frames, 960×540, 64 samples), the full pipeline takes approximately 6 hours per scene, including all object- and dynamic-reflection iterations, VLM feedback calls, and Blender rendering. The average API cost is $1.23 per scene using Gemini 3. We will include these runtime and cost statistics in the revised manuscript; more detailed end-to-end generation details are also provided in our response to **reviewer e4Ub, Regarding W2 and Q1**.
>
> ### Regarding Limitation
> We thank the reviewer for this important point. Our method is a general language-to-simulation pipeline rather than one inherently restricted to a specific catalog. It also supports a three-stage graceful-degradation mechanism for vocabulary coverage, including local retrieval, external asset retrieval, and direct LLM generation.

---

> > ### Author Rebuttal · Reviewer_g5tr · 2026-04-03
> >
> > I have read the rebuttal and my main concerns are now largely resolved. The authors clarified that Code4D contains about 50 prompts rather than only the 10 shown in Table 6, added human evaluation supporting the GPT-based metrics, reported runtime and cost statistics, and provided a fairer semantic/physical comparison against video baselines. Based on these clarifications, I believe the paper is stronger than my initial assessment, and I am raising my score accordingly.

---

> > > ### Author Response · Authors · 2026-04-04
> > >
> > > Thank you for your acknowledgement and support. We sincerely appreciate your time, careful consideration, and thoughtful feedback throughout the review process. We are grateful that our rebuttal helped clarify the key points and address your main concerns.

---

### Official Review · Reviewer_paSK · 2026-03-12

**Soundness:** 2
**Presentation:** 3
**Significance:** 2
**Originality:** 2
**Overall Recommendation:** 5
**Confidence:** 2

**Summary:**

This paper presents Code2Worlds, a language-to-simulation framework for generating 4D dynamic worlds as executable code rather than static 3D scenes. Specifically, it introduces a dual-stream architecture that separates retrieval-augmented object generation from hierarchical environment construction, and a physics-aware closed-loop refinement pipeline in which a post-processing agent adds motion and a VLM-based motion critic iteratively corrects dynamic errors. The paper also present benchmark for evaluating 4D code-based world generation and reports improved scene quality, richness, and dynamic consistency over prior baselines.

**Compliance With Llm Reviewing Policy:**

Affirmed.

**Final Justification:**

After rebuttal, I’d like to raise my ratting for acceptance.

**Key Questions For Authors:**

I would be interested in seeing more detail on the practical end-to-end cost of the full pipeline, including VLM usage cost, overall generation success rate, and common failure modes. It would also be helpful to better understand the method’s scalability: for example, how large or complex the generated scenes can be, and what range of motions or dynamic interactions the system can reliably support. In addition, stronger qualitative comparisons with baselines would make it easier to assess.

The ablation results suggest that retrieval is one of the most critical components, as removing it leads to the largest drop in object-generation quality (e.g., SGS). It would be useful to include more controlled and fair comparisons per-module, that better clarify why each design choice is preferable, and whether there are trade-offs among alternatives.

**Limitations:**

Yes. The paper does discuss limitations, primarily from the perspective of computational bottlenecks. It would be even stronger if the discussion also covered the current quality ceiling of the approach, for example, limitations in visual realism, scene complexity, motion fidelity, and physical consistency, as these are likely to be equally important directions for future improvement.

**Strengths And Weaknesses:**

The overall system design is coherent and reasonably well motivated: the method adopts a dual-stream architecture that separates object generation from scene construction, followed by a physics-aware post-processing and self-refinement loop. The high-level decomposition is easy to follow. The supplementary videos are also helpful to assess the generation quality.

That said, the contribution feels more like a system-level composition of multiple stage-wise components than a sharply new technical advance. While I do see practical value in the paper and agree that the problem is relevant, the significance and originality currently feel moderate. Code-based world generation is a promising and potentially broad direction, and the paper would benefit from stating more clearly what application domain or use setting it is specifically targeting, beyond simply framing the task as 4D generation. The supplemented visual results still retain a somewhat low-poly or unrealistic appearance/dynamics, and the paper would be stronger with direct qualitative comparisons against baseline methods.

---

> ### Author Rebuttal · Authors · 2026-03-31
>
> ## Response to Reviewer paSK
>
> We thank the reviewer for the constructive feedback and important questions.
>
> *"That said, the contribution feels more like [...]"*
>
> We thank the reviewer for this helpful suggestion. We agree that the target use setting should be stated more explicitly. While our work is system-oriented, its contribution is not a generic stage-wise composition, but a 4D-specific language-to-simulation framework with a dual-stream design and physics-aware closed-loop correction. Our primary application is the generation of physically consistent and controllable 4D simulations for embodied AI and sim-to-real transfer, where executable, editable, and physically grounded environments are essential. In this sense, our goal is not generic 4D content creation, but simulation-ready world generation. We will revise the introduction and discussion to clarify the application scope.
>
> *"The supplemented visual results still [...] I would be interested in seeing more detail [...]"*
>
> We agree that practical cost and failure modes should be stated more clearly. In our current implementation, the average API cost is $1.23 per scene using Gemini 3; more detailed runtime statistics are provided in our response to **Reviewer e4Ub, Regarding W2 and Q1**.
>
> We will also report failure cases more explicitly. The main failures are dynamic-control errors and scene-complexity failures in large or cluttered scenes. Our current operating regime is best characterized as medium-complexity, simulation-ready scenes with parameterizable physical dynamics, rather than arbitrary open-vocabulary world generation. It is currently most reliable for gravity- or force-driven motion, collision-aware actuation, and particle or water effects, while long-horizon multi-object interactions remain challenging.
>
> We also agree that more direct qualitative baseline comparisons would strengthen the paper. Our qualitative advantage is mainly in finer object-level semantic grounding, richer environmental composition, and more physically plausible motion. For example, our method better captures compositional object attributes (e.g., curved, withered leaves with brown spots), produces denser, more semantically aligned ecological scenes, and exhibits physically grounded motion, such as rolling, rather than implausible translation. We will include clearer side-by-side comparisons in the revision.
>
>
> *"The ablation results suggest that retrieval is [...]"*
>
> We thank the reviewer for this helpful comment. To better explain the role of each module and the associated trade-offs, we added two additional controlled analyses.
>
> (a) Retrieval variants
> | Retrieval Variant       | O-CLIP | SGS | Style-CLIP |
> |-------------------------|---------:|------:|-------------:|
> | w/o Retrieve            | 0.2221   | 23.5  | 0.6578       |
> | Random Retrieve         | 0.2209   | 25.1  | 0.6673       |
> | Category-only Retrieve  | 0.2301   | 26.7  | 0.6539       |
> | Ours | 0.2655 | 61.4 | 0.6734 |
>
> First, we tested weaker retrieval alternatives to clarify why retrieval matters. Both random retrieval and category-only retrieval perform worse than our full-reference script retrieval. This shows that the gain does not come from retrieval in the abstract or from coarse category hints, but from retrieving semantically matched reference scripts.
>
> (b) Quality–cost trade-off
> | Variant          | O-CLIP | SGS | Avg. Obj. Iters | API Cost ($) |
> |------------------|---------:|------:|------------------:|--------------------:|
> | w/o Retrieve     | 0.2221   | 23.5  | 5.0               | 0.23                |
> | w/o Lparam       | 0.2511   | 48.8  | 4.0               | 0.26                |
> | w/o VLM-Critic   | 0.2388   | 58.6  | N/A               | 0.06                |
> | Ours         | 0.2655 | 61.4 | 2.5         | 0.17                |
>
> Second, we added a quality–cost trade-off analysis. The results indicate complementary roles rather than redundancy. Retrieval mainly improves initialization quality: removing it sharply degrades object fidelity, increases the number of iterations, and increases API cost. Lparam mainly improves attribute grounding: removing it also lowers quality and increases cost. VLM-Critic mainly acts as a refinement module: removing it results in a quality drop while reducing API costs.
>
> Overall, these results clarify that our modules address different bottlenecks in the pipeline rather than functioning as redundant add-ons.
>
> *"Yes. The paper does discuss [...]"*
> We thank the reviewer for this helpful comment. The current method is limited by Infinigen's realism ceiling and is most reliable for medium-complexity scenes. Motion fidelity and physical consistency will degrade in dense interactions, severe occlusion, and long-horizon dynamics.

---

> > ### Author Rebuttal · Reviewer_paSK · 2026-04-03
> >
> > Thanks for the explanation. Most of my main questions have been addressed.

---

> > > ### Author Response · Authors · 2026-04-04
> > >
> > > Thank you for your acknowledgement and support. We greatly appreciate your time, thoughtful review, and careful consideration throughout the review process. We are pleased that our rebuttal was able to clarify the key issues and address your main concerns.

---

### Official Review · Reviewer_e4Ub · 2026-03-12

**Soundness:** 3
**Presentation:** 3
**Significance:** 3
**Originality:** 3
**Overall Recommendation:** 5
**Confidence:** 3

**Summary:**

This paper introduces Code2Worlds, a framework that formulates 4D scene generation as a language to simulation code generation problem. The key idea is a dual-stream architecture: an Object Stream that uses retrieval-augmented generation to produce high-fidelity 3D objects via Infinigen's procedural generator, and a Scene Stream that hierarchically decomposes environment instructions into parametric specifications. A PostProcess Agent integrates these streams and scripts physics dynamics in Blender, while a VLM-Motion component provides closed-loop feedback to iteratively correct physical plausibility. The authors also introduce Code4D, a benchmark for evaluating 4D scene generation. Experiments show substantial improvements: 41% higher SGS and 49% higher Richness than prior code-to-scene methods, along with superior temporal consistency over video diffusion baselines.

**Compliance With Llm Reviewing Policy:**

Affirmed.

**Key Questions For Authors:**

1. How many reflection iterations does the system typically need before convergence (for both object and dynamic loops)? What is the total wall-clock time and LLM API cost per scene? This is critical for understanding the practical applicability of the approach.

2. How does the framework handle instructions that require objects or environments outside Infinigen's procedural vocabulary? For instance, "a car driving through a rainy parking lot" would require man-made assets. Is there a graceful degradation or fallback mechanism?

**Limitations:**

yes

**Strengths And Weaknesses:**

Strengths:
- The core insight is sound: decomposing 4D generation into object-level detail and scene-level data addresses a real limitation of previous approaches that struggle to maintain local fidelity and global coherence.
- The physics-aware closed-loop correction via VLM-Motion feedback is a practical design choice. The ablation data shows its value clearly: removing VLM-Motion causes physics failure rate to increase significantly..
- The paper evaluates across multiple metrics (CLIP variants, VBench, GPT-4o-based SGS/HRS/Richness, and a human-inspected physics failure rate), and compares against both code-centric and video diffusion baselines.
- The ablation study is well-designed, systematically removing each component (retrieval, parameter library, VLM-Critic, VLM-Motion) and showing clear degradation, which validates the design choices.
- The qualitative results show genuinely impressive physical effects (relighting, water spill, fire) that prior methods cannot produce.



Weaknesses:
- The evaluation relies heavily on GPT-4o as a judge for several key metrics. These metrics have known biases and the paper does not report inter-rater agreement with human evaluators. A human study, even small-scale, would significantly strengthen the claims.
- The paper doesn't discuss computational cost or latency. The iterative self-reflection loops (both object-level and dynamic-level) involve multiple LLM calls, rendering passes, and VLM evaluations per scene. How many iterations does convergence typically require? What is the wall-clock time per scene compared to baselines?
- The parameter retrieval step requires a Procedural Parameters Library and Reference Code Library constructed from Infinigen scripts. The effort to build and maintain these is not discussed, and scalability to new object categories is unclear.

---

> ### Author Rebuttal · Authors · 2026-03-31
>
> ## Response to Reviewer e4Ub
>
> We thank the reviewer for raising these important points.
>
> ### Regarding W1: reliance on GPT-based evaluation
>  | Method       | SGS  | Richness | Human SGS | Human Richness |
> |--------------|-----:|---------:|----------:|---------------:|
> | MeshCoder    | 14.6 | -        | 1.16      | -              |
> | Infinigen    | 35.5 | 41.0     | 2.33      | 2.80           |
> | 3D-GPT       | 37.0 | 41.7     | 1.66      | 2.30           |
> | SceneCraft   | 34.6 | 15.2     | 1.50      | 1.20           |
> | ImmerseGen   | 43.5 | 35.5     | 2.66      | 2.90           |
> | BlenderMCP   | 45.8 | 32.0     | 2.83      | 2.20           |
> | BlenderGPT   | 35.1 | 32.8     | 2.50      | 2.10           |
> | Code2Worlds  | 61.4 | 62.3     | 4.33      | 4.00           |
>
> | Method       | HRS  | Human HRS |
> |--------------|-----:|---------:|
> | w/o VLM-Motion    | 47 |    2.2   |
> | Code2Worlds    | 55.4 |   3.6   |
>
> | Metric    | Spearman ρ | p-value | Pearson r | p-value |
> |-----------|-----------:|--------:|----------:|--------:|
> | SGS       | 0.905      | 0.0020  | 0.906     | 0.0019  |
> | Richness  | 0.821      | 0.0234  | 0.944     | 0.0014  |
>
> We thank the reviewer for raising this important concern. To directly address it, we conducted an additional human evaluation on SGS, Richness, and HRS, using 1–5 ratings. The human results are strongly consistent with the GPT-based metrics in both ranking and correlation analysis: SGS and Richness show strong correlation with human judgment, and Code2Worlds ranks first in both Human SGS (4.33) and Human Richness (4.00). For dynamic physical plausibility, human raters also prefer Code2Worlds over w/o VLM-Motion (3.6 vs. 2.2), consistent with the GPT-HRS trend (55.4 vs. 47.0). These results show that SGS, Richness, and HRS are not arbitrary self-referential scores, but meaningful perceptual proxies aligned with human judgment.
>
> Therefore, our conclusions are not based on GPT-4o scores alone, nor are SGS / Richness / HRS used as self-referential stand-ins. Rather, they serve as supplementary perceptual metrics whose validity is now supported by human evaluation. We will add the full human-evaluation protocol and agreement statistics.
>
> ### Regarding W2 and Q1: computational cost, latency, and convergence
> We thank the reviewer for raising this important point. We agree that computational cost and latency are important for assessing the practical applicability of our system. In our implementation, the object-level reflection loop converges in 2.5 iterations on average, while the dynamic-level reflection loop converges in 4.5 iterations on average. To keep the process bounded in practice, we use hard iteration caps of 4 rounds for object reflection and 6 rounds for dynamic reflection. Under this rendering setup (single RTX 4090, 120 frames, 960×540 resolution, 64 samples), the average wall-clock time is approximately 6 hours per scene. The average LLM/VLM API cost is $1.23 per scene, using Gemini 3. We will add these convergence statistics, runtime details, and API cost numbers to the revised manuscript for better transparency and reproducibility.
>
> ### Regarding W3 and Q2: scalability to object categories
> We thank the reviewer for raising this important question. Our framework should be viewed as a general language-to-simulation pipeline, rather than a method inherently restricted to a specific asset catalog. The current experiments are instantiated on Infinigen’s factory catalog, since it provides a convenient procedural interface and executable physical controls, but the overall design is not tied to Infinigen itself. The same pipeline can be extended to other catalogs or asset sources by adapting the retrieval index and keeping a unified downstream object interface.
>
> More concretely, our framework supports a three-stage graceful-degradation mechanism for vocabulary coverage: retrieval from the local knowledge base, retrieval from external 3D asset repositories, and direct LLM generation when retrieval is unavailable. For the reviewer’s example, “a car driving through a rainy parking lot,” the rainy environment and terrain are handled by the Scene Stream, while the car can be sourced through this staged object pathway. Since all stages produce a unified object representation, the downstream PostProcess Stream and physics-aware closed-loop refinement can be applied in the same way regardless of object source. Thus, the framework degrades gracefully rather than failing hard on out-of-vocabulary objects. We will clarify both this generality and the current experimental scope more explicitly in the revision.

---

> > ### Author Rebuttal · Reviewer_e4Ub · 2026-04-06
> >
> > The authors successfully addressed all of my points in their rebuttal. Thank you.

---

> > > ### Author Response · Authors · 2026-04-07
> > >
> > > Thank you for your acknowledgement and support. We sincerely appreciate your time, constructive feedback, and careful consideration throughout the review process. We are pleased that our rebuttal was able to clarify the key points and address your main concerns.

---

### Official Review · Reviewer_vqnP · 2026-03-13

**Soundness:** 2
**Presentation:** 3
**Significance:** 3
**Originality:** 3
**Overall Recommendation:** 4
**Confidence:** 5

**Summary:**

Code2Worlds proposes a dual-stream multi-agent framework for 4D scene generation via coding LLMs, combining RAG-based parametric object generation over Infinigen with a VLM-Motion Critic for closed-loop physics refinement.

**Compliance With Llm Reviewing Policy:**

Affirmed.

**Key Questions For Authors:**

Q1. Given that Idea23D (COLING 2025) already implements a multi-agent closed-loop with VLM feedback reflection for 3D generation, and that Blender-MCP already supports interactive VLM-driven Blender code generation, what is the irreducible scientific novelty of Code2Worlds?

Q2. Why are Blender-MCP and BlenderGPT not included as baselines? A quantitative comparison on Code4D against these open-source tools is necessary to establish that Code2Worlds offers any meaningful advance.

Q3. With only 10 prompts and author-defined GPT-4o metrics, the evaluation is insufficient. Why was human evaluation not conducted, and how do the authors validate that SGS/HRS/Richness are meaningful measures?

Q4: The authors claim Code2Worlds is the first to generate 4D environments via code generation. However, it remains unclear how the VLM reliably understands and generates per-instance motion in complex dynamic scenes. Based on the reviewer's own reproduction of VIGA (Vision-as-Inverse-Graphics Agent via Interleaved Multimodal Reasoning), existing VLMs including Qwen, Claude, and GPT-4o generally fail at 4D scene generation tasks of this complexity. The authors should provide a rigorous failure case analysis and clarify under what conditions the VLM-Motion Critic produces physically correct motion code, rather than relying solely on success cases in the qualitative results.

**Limitations:**

yes

**Strengths And Weaknesses:**

# Strengths

1. The VLM-Motion Critic for temporal physics correction is a concrete and useful contribution.
2. Ablation study (Tables 3–4) clearly supports the design choices.

# Weaknesses

1. First, the core paradigm of multi-agent coordination + code generation + VLM closed-loop reflection for 3D AIGC has been thoroughly explored in prior work. The following directly predate or overlap with this paper's central claims:

[1] CCA: Collaborative Competitive Agents for Image Editing, FCS

[2] SceneCraft: An LLM agent for synthesizing 3d scenes as Blender code. ICML, 2024.

[3] https://github.com/gd3kr/BlenderGPT

[4] 3D-GPT: Procedural 3D Modeling with Large Language Models, 3DV, 2025

[5] Idea23D: Collaborative LMM Agents Enable 3D Model Generation from Interleaved Multimodal Inputs, COLING 25, a three-agent closed-loop (generation → selection → feedback reflection) for 3D content, architecturally nearly identical to the Object Stream + VLM-Critic loop proposed here.

[6] LL3M: Large Language 3D Modelers, a multi-agent system using Blender Python code for 3D asset generation.

[7] Vision-as-Inverse-Graphics Agent via Interleaved Multimodal Reasoning

Second, VLM-driven Blender code generation is already a solved engineering problem in the open-source community. Blender-MCP (github.com/ahujasid/blender-mcp, github.com/CommonSenseMachines/blender-mcp) and BlenderGPT (github.com/gd3kr/BlenderGPT) provide mature, freely available pipelines for LLM/VLM-guided Blender scene creation. In the reviewer's experience, these tools already achieve effects qualitatively comparable to what is demonstrated in this paper. The authors neither cite nor benchmark against any of these.

Taken together, Code2Worlds reads as an engineering integration of existing multi-agent patterns onto Infinigen's APIs, not a scientific contribution.

2. The Code4D benchmark has only 10 prompts. The primary metrics (SGS, HRS, Richness) are GPT-4o-based scores defined by the authors with no human evaluation or validation. This self-referential evaluation design is a soundness concern.

---

> ### Author Rebuttal · Authors · 2026-03-31
>
> ## Response to Reviewer vqnP
>
> We thank the reviewer for the thoughtful and constructive feedback.
>
> ### Regarding W1 and Q1: novelty beyond prior multi-agent 3D systems
> We thank the reviewer for this important comment. We agree that prior works have explored key aspects of agentic generation and visual feedback, and we will cite them in the revised version. Specifically, CCA studies multi-agent reflection for image editing; SceneCraft formulates text-to-3D scene synthesis as Blender code with refinement; BlenderGPT and Blender-MCP enable language-driven Blender control and code-based 3D creation; 3D-GPT performs instruction-driven procedural 3D modeling; Idea23D adopts a three-agent self-refinement loop for multimodal IDEA-to-3D generation; LL3M uses multi-agent Blender code generation for 3D asset creation and editing; and VIGA introduces a write-run-render-compare-revise loop for scene reconstruction and editing.
>
> However, our novelty claim is not the generic outer loop itself, but its specialization to language-to-simulation 4D world generation. Unlike prior works on image editing, static 3D asset and scene generation, or inverse-graphics reconstruction, Code2Worlds targets physically grounded 4D environments from text. In particular, Idea23D operates within a T2I + I2-3D pipeline for static 3D model generation, while Blender-MCP focuses on interactive Blender control; by contrast, our setting requires not only visual alignment, but also executable, temporally and physically consistent dynamics. This is reflected in our architecture, which addresses two 4D-specific bottlenecks: multi-scale context entanglement, motivating the dual-stream object and scene factorization, and the semantic-physical execution gap, motivating the physics-aware VLM-Motion closed loop. In this sense, our contribution is a 4D-specialized language-to-simulation framework for controllable and physically plausible world generation.
>
> ### Regarding W1 and Q2: Blender-MCP and BlenderGPT as baselines
> | Method        | O-CLIP | SGS | Style-CLIP | S-CLIP | Richness |
> |---------------|----------:|------:|-------------:|---------:|-----------:|
> | Blender-MCP   |    0.2563    |  45.8  |      0.6476     |   0.2398    |    32.0     |
> | BlenderGPT    |    0.2498    |  35.1  |      0.6331     |   0.2428    |    32.8     |
> | Code2Worlds          |   0.2655  | 61.4  |    0.6734    |  0.2432  |    62.3    |
>
> We thank the reviewer for this important suggestion. To address this, we reproduced Blender-MCP and BlenderGPT using the same Code4D prompts and added a direct quantitative comparison of static object and scene metrics. Code2Worlds consistently outperforms both baselines across all reported metrics, with especially large gains on SGS and Richness. This suggests that our method provides markedly better fine-grained object fidelity and richer scene construction, rather than only marginal improvements over existing open-source Blender pipelines. We will include these baselines in the revision.
>
> ### Regarding W2 and Q3: human evaluation and validation of GPT-based metrics
> We thank the reviewer for raising this important concern. We refer the reviewer to our response to **Reviewer e4Ub, Regarding W1**, for the full human-evaluation protocol, correlation statistics, and detailed results. Briefly, we conducted an additional human evaluation on SGS, Richness, and HRS. The human results are strongly consistent with the GPT-based metrics in both overall ranking and correlation analysis, and support the same conclusion.
>
> ### Regarding Q4: reliability and failure cases of VLM-Motion
> We thank the reviewer for this important comment. We would like to clarify a key point: VLM-Motion does not directly generate motion code. Instead, it inspects rendered rollouts and provides corrective feedback, while code revision is performed by the coding LLM within our constrained simulation pipeline. Our claim is therefore not that current VLMs can one-shot generate reliable 4D motion for arbitrary complex scenes, but that VLM feedback becomes effective inside a parameterized, executable physics loop. This is also why our setting is more specialized than a task-agnostic agent such as VIGA. Empirically, this module is necessary: removing VLM-Motion increases Failure Rate from 10% to 60% and reduces HRS from 55.4 to 47.0.
>
> It is most reliable when scene geometry is already well grounded, the motion can be expressed through structured physical controls, and the error is visible from a short rollout. It is less reliable in crowded scenes, under heavy occlusion, or in long-horizon many-body interactions. A representative failure case is dense leaf motion under changing wind and collision conditions, where the critic may detect that the overall motion is unnatural but fail to localize the exact cause, leading to synchronized motion, interpenetration, hovering leaves, or abrupt settling. We will add such representative failure cases and clarify this scope in the revision.

---

> > ### Author Rebuttal · Reviewer_vqnP · 2026-04-03
> >
> > Thank you for the detailed response. After carefully considering the comments from other reviewers, I maintain my original score.

---

> > > ### Author Response · Authors · 2026-04-04
> > >
> > > Thank you for your acknowledgement and support. We truly appreciate your time, constructive feedback, and thoughtful consideration throughout the review process. We are glad that our rebuttal helped clarify the main points and resolve your primary concerns.

---

### Decision · Program_Chairs · 2026-04-30

**Decision:**

Accept (regular)

**Comment:**

This paper proposes Code2Worlds, a language-to-simulation framework for generating physically grounded 4D scenes via code, combining a dual-stream object/scene generation pipeline with a VLM-based motion critic for closed-loop physics refinement.

The reviewers agree that the problem is timely and important, and they find the overall system design coherent and practically compelling, with the VLM-Motion refinement loop standing out as the most interesting contribution. The initial discussion raised concerns about limited novelty relative to prior agentic code-generation systems, heavy reliance on GPT-based evaluation, unclear benchmark scale, runtime/cost, and the scope restriction imposed by Infinigen-style procedural assets. The rebuttal substantially strengthened the paper by adding comparisons to Blender-MCP and BlenderGPT, providing human evaluation and correlation analysis supporting the GPT-based metrics, clarifying that Code4D contains roughly 50 prompts, reporting runtime and API cost, and giving a fairer semantic/physical comparison against video baselines. Some concerns remain that the paper is more of a strong system integration than a sharply new technical advance, and that realism, scene complexity, and open-vocabulary coverage are still limited by the current asset and simulation stack.

Overall, however, the rebuttal addressed the main soundness concerns well enough that reviewer confidence increased and the final consensus is clearly positive. Given this, the recommendation is Accept.